# Learner-aware Teaching: Inverse Reinforcement Learning with Preferences and Constraints

**Sebastian Tschiatschek**[*]
Microsoft Research
setschia@microsoft.com

**Ahana Ghosh**[*]
MPI-SWS
gahana@mpi-sws.org

**Luis Haug**[*]
ETH Zurich
lhaug@inf.ethz.ch

**Rati Devidze**
MPI-SWS
rdevidze@mpi-sws.org

**Adish Singla**
MPI-SWS
adishs@mpi-sws.org

## Abstract

Inverse reinforcement learning (IRL) enables an agent to learn complex behavior by observing demonstrations from a (near-)optimal policy. The typical assumption is that the learner's goal is to match the teacher's demonstrated behavior. In this paper, we consider the setting where the learner has its own preferences that it additionally takes into consideration. These preferences can for example capture behavioral biases, mismatched worldviews, or physical constraints. We study two teaching approaches: *learner-agnostic* teaching, where the teacher provides demonstrations from an optimal policy ignoring the learner's preferences, and *learner-aware* teaching, where the teacher accounts for the learner's preferences. We design learner-aware teaching algorithms and show that significant performance improvements can be achieved over learner-agnostic teaching.

## 1 Introduction

Inverse reinforcement learning (IRL) enables a learning agent (*learner*) to acquire skills from observations of a *teacher*'s demonstrations. The learner infers a reward function explaining the demonstrated behavior and optimizes its own behavior accordingly. IRL has been studied extensively [Abbeel and Ng, 2004, Ratliff et al., 2006, Ziebart, 2010, Boularias et al., 2011, Osa et al., 2018] under the premise that the learner can and is willing to imitate the teacher's behavior.

In real-world settings, however, a learner typically does not blindly follow the teacher's demonstrations, but also has its own preferences and constraints. For instance, consider demonstrating to an auto-pilot of a self-driving car how to navigate from A to B by taking the most fuel-efficient route. These demonstrations might conflict with the preference of the auto-pilot to drive on highways in order to ensure maximum safety. Similarly, in robot-human interaction with the goal of teaching people how to cook, a teaching robot might demonstrate to a human user how to cook "roast chicken", which could conflict with the preferences of the learner who is "vegetarian". To give yet another example, consider a surgical training simulator which provides virtual demonstrations of expert behavior; a novice learner might not be confident enough to imitate a difficult procedure because of safety concerns. In all these examples, the learner might not be able to acquire useful skills from the teacher's demonstrations.

In this paper, we formalize the problem of teaching a learner with preferences and constraints. First, we are interested in understanding the suboptimality of *learner-agnostic* teaching, i.e., ignoring the learner's preferences. Second, we are interested in designing *learner-aware* teachers who account

---

[*]Authors contributed equally to this work.

for the learner's preferences and thus enable more efficient learning. To this end, we study a learner model with preferences and constraints in the context of the Maximum Causal Entropy (MCE) IRL framework [Ziebart, 2010, Ziebart et al., 2013, Zhou et al., 2018]. This enables us to formulate the teaching problem as an optimization problem, and to derive and analyze algorithms for learner-aware teaching. Our main contributions are:

I   We formalize the problem of IRL under preference constraints (Section 2 and Section 3).

II  We analyze the problem of optimizing demonstrations for the learner when preferences are *known* to the teacher, and we propose a bilevel optimization approach to the problem (Section 4).

III We propose strategies for adaptively teaching a learner with preferences *unknown* to the teacher, and we provide theoretical guarantees under natural assumptions (Section 5).

IV  We empirically show that significant performance improvements can be achieved by learner-aware teachers as compared to learner-agnostic teachers (Section 6).

## 2   Problem Setting

**Environment.** Our environment is described by a *Markov decision process* (MDP) $\mathcal{M} := (\mathcal{S}, \mathcal{A}, T, \gamma, P_0, R)$. Here $\mathcal{S}$ and $\mathcal{A}$ denote finite sets of states and actions. $T \colon \mathcal{S} \times \mathcal{S} \times \mathcal{A} \to [0,1]$ describes the state transition dynamics, i.e., $T(s'|s,a)$ is the probability of landing in state $s'$ by taking action $a$ from state $s$. $\gamma \in (0,1)$ is the discounting factor. $P_0 \colon \mathcal{S} \to [0,1]$ is an initial distribution over states. $R \colon \mathcal{S} \to \mathbb{R}$ is the reward function. We assume that there exists a feature map $\phi_r \colon \mathcal{S} \to [0,1]^{d_r}$ such that the reward function is linear, i.e., $R(s) = \langle \mathbf{w}_r^*, \phi_r(s) \rangle$ for some $\mathbf{w}_r^* \in \mathbb{R}^{d_r}$. Note that a bound of $\|\mathbf{w}_r^*\|_1 \leq 1$ ensures that $|R(s)| \leq 1$ for all $s$.

**Basic definitions.** A *policy* is a map $\pi \colon \mathcal{S} \times \mathcal{A} \to [0,1]$ such that $\pi(\,\cdot\,|\,s)$ is a probability distribution over actions for every state $s$. We denote by $\Pi$ the set of all such policies. The performance measure for policies we are interested in is the *expected discounted reward* $R(\pi) := \mathbb{E}\left(\sum_{t=0}^{\infty} \gamma^t R(s_t)\right)$, where the expectation is taken with respect to the distribution over trajectories $\xi = (s_0, s_1, s_2, \ldots)$ induced by $\pi$ together with the transition probabilities $T$ and the initial state distribution $P_0$. A policy $\pi$ is *optimal* for the reward function $R$ if $\pi \in \arg\max_{\pi' \in \Pi} R(\pi')$, and we denote an optimal policy by $\pi^*$. Note that $R(\pi) = \langle \mathbf{w}_r^*, \mu_r(\pi) \rangle$, where $\mu_r \colon \Pi \to \mathbb{R}^{d_r}, \pi \mapsto \mathbb{E}\left(\sum_{t=0}^{\infty} \gamma^t \phi_r(s_t)\right)$, is the map taking a policy to its vector of *(discounted) feature expectations*. We denote by $\Omega_r = \{\mu_r(\pi) : \pi \in \Pi\}$ the image $\mu_r(\Pi)$ of this map. Note that the set $\Omega_r \in \mathbb{R}^{d_r}$ is convex (see [Ziebart, 2010, Theorem 2.8] and [Abbeel and Ng, 2004]), and also bounded due to the discounting factor $\gamma \in (0,1)$. For a finite collection of trajectories $\Xi = \{s_0^i, s_1^i, s_2^i, \ldots\}_{i=1,2,\ldots}$ obtained by executing a policy $\pi$ in the MDP $\mathcal{M}$, we denote the empirical counterpart of $\mu_r(\pi)$ by $\hat{\mu}_r(\Xi) := \frac{1}{|\Xi|} \sum_i \sum_t \gamma^t \phi_r(s_t^i)$.

**An IRL learner and a teacher.** We consider a learner $\mathsf{L}$ implementing an inverse reinforcement learning (IRL) algorithm and a teacher $\mathsf{T}$. The teacher has access to the full MDP $\mathcal{M}$; the learner knows the MDP and the parametric form of reward function $R(s) = \langle \mathbf{w}_r, \phi_r(s) \rangle$ but does not know the true reward parameter $\mathbf{w}_r^*$. The learner, upon receiving demonstrations from the teacher, outputs a policy $\pi^{\mathsf{L}}$ using its algorithm. The teacher's objective is to provide a set of demonstrations $\Xi^{\mathsf{T}}$ to the learner that ensures that the learner's output policy $\pi^{\mathsf{L}}$ achieves high reward $R(\pi^{\mathsf{L}})$.

The standard IRL algorithms are based on the idea of *feature matching* [Abbeel and Ng, 2004, Ziebart, 2010, Osa et al., 2018]: The learner's algorithm finds a policy $\pi^{\mathsf{L}}$ that matches the feature expectations of the received demonstrations, ensuring that $\|\mu_r(\pi^{\mathsf{L}}) - \hat{\mu}_r(\Xi^{\mathsf{T}})\|_2 \leq \epsilon$ where $\epsilon$ specifies a desired level of accuracy. In this standard setting, the learner's primary goal is to imitate the teacher (via feature matching) and this makes the teaching process easy. In fact, the teacher just needs to provide a sufficiently rich pool of demonstrations $\Xi^{\mathsf{T}}$ obtained by executing $\pi^*$, ensuring $\|\hat{\mu}_r(\Xi^{\mathsf{T}}) - \mu_r(\pi^*)\|_2 \leq \epsilon$. This guarantees that $\|\mu_r(\pi^{\mathsf{L}}) - \mu_r(\pi^*)\|_2 \leq 2\epsilon$. Furthermore, the linearity of rewards and $\|\mathbf{w}_r^*\|_1 \leq 1$ ensures that the learner's output policy $\pi^{\mathsf{L}}$ satisfies $R(\pi^{\mathsf{L}}) \geq R(\pi^*) - 2\epsilon$.

**Key challenges in teaching a learner with preference constraints.** In this paper, we study a novel setting where the learner has its own preferences which it additionally takes into consideration when learning a policy $\pi^{\mathsf{L}}$ using teacher's demonstrations. We formally specify our learner model in the next section; here we highlight the key challenges that arise in teaching such a learner. Given that the learner's primary goal is no longer just imitating the teacher via feature matching, the learner's output policy can be suboptimal with respect to the true reward even if it had access to $\mu_r(\pi^*)$, i.e.,

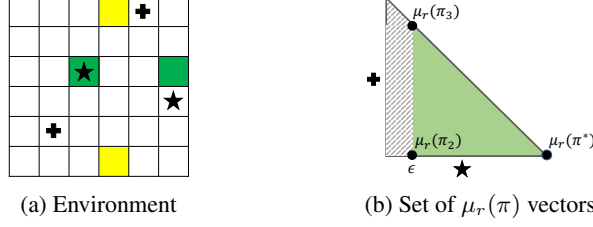

(a) Environment          (b) Set of $\mu_r(\pi)$ vectors

Figure 1: An illustrative example to showcase the suboptimality of teaching when the learner has preferences and constraints. **Environment:** Figure 1a shows a grid-world environment inspired by the object-world and gathering game environments [Levine et al., 2010, Leibo et al., 2017, Mendez et al., 2018]. Each cell represents a state, there are five actions given by "left", "up", "right", "down", "stay", the transitions are deterministic, and the starting state is the top-left cell. The agent's goal is to collect objects in the environment: Collecting a "star" provides a reward of $1.0$ and a "plus" a reward of $0.9$; objects immediately appear again upon collection, and the rewards are discounted with $\gamma$ close to $1$. The optimal policy $\pi^*$ is to go to the nearest "star" and then "stay" there. **Preferences:** A small number of states in the environment are distractors, depicted by colored cells in Figure 1a. We consider a learner who prefers to avoid "green" distractors: it has a hard constraint that the probability of having a "green" distractor within a 3x3 neighborhood, i.e., 1-cell distance, is at most $\epsilon = 0.1$. **Feature expectation vectors:** Figure 1b shows the set of feature expectation vectors $\{\mu_r(\pi) : \pi \in \Pi\}$. The $x$-axis and the $y$-axis represent the discounted feature count for collecting "star" and "plus" objects, respectively. The striped region represents policies that are feasible w.r.t. the learner's constraint. **Suboptimality of teaching:** Upon receiving demonstrations from an optimal policy $\pi^*$ with feature vector $\mu_r(\pi^*)$, the learner under its preference constraint can best match the teacher's demonstrations (in a sense of minimizing $\|\mu_r(\pi^L) - \mu_r(\pi^*)\|_2$) by outputting a policy with $\mu_r(\pi_2)$, which is clearly suboptimal w.r.t. the true rewards. Policy $\pi_3$ with feature vector $\mu_r(\pi_3)$ represents an alternate teaching policy which would have led to higher reward for the learner.

the feature expectation vector of an optimal policy $\pi^*$. Figure 1 provides an illustrative example to showcase the suboptimality of teaching when the learner has preferences and constraints. The key challenge that we address in this paper is that of designing a teaching algorithm that selects demonstrations while accounting for the learner's preferences.

## 3 Learner Model

In this section we describe the learner models we consider, including different ways of defining preferences and constraints. First, we introduce some notation and definitions that will be helpful. We capture learner's preferences via a feature map $\phi_c : \mathcal{S} \to [0,1]^{d_c}$. We define $\phi(s)$ as a concatenation of the two feature maps $\phi_r(s)$ and $\phi_c(s)$ given by $[\phi_r(s)^\dagger, \phi_c(s)^\dagger]^\dagger$ and let $d = d_r + d_c$. Similar to the map $\mu_r$, we define $\mu_c \colon \Pi \to \mathbb{R}^{d_c}$, $\pi \mapsto \mathbb{E}\left(\sum_{t=0}^{\infty} \gamma^t \phi_c(s_t)\right)$ and $\mu \colon \Pi \to \mathbb{R}^d$, $\pi \mapsto \mathbb{E}\left(\sum_{t=0}^{\infty} \gamma^t \phi(s_t)\right)$. Similar to $\Omega_r$, we define $\Omega_c \subseteq \mathbb{R}^{d_c}$ and $\Omega \subseteq \mathbb{R}^d$ as the images of the maps $\mu_c(\Pi)$ and $\mu(\Pi)$. Note that for any policy $\pi \in \Pi$, we have $\mu(\pi) = [\mu_r(\pi)^\dagger, \mu_c(\pi)^\dagger]^\dagger$.

**Standard (discounted) MCE-IRL.** Our learner models build on the (discounted) Maximum Causal Entropy (MCE) IRL framework [Ziebart et al., 2008, Ziebart, 2010, Ziebart et al., 2013, Zhou et al., 2018]. In the standard (discounted) MCE-IRL framework, a learning agent aims to identify a policy that matches the feature expectations of the teacher's demonstrations while simultaneously maximizing the (discounted) causal entropy given by $H(\pi) := H(\{a_t\}_{t=0,1,\dots} \| \{s_t\}_{t=0,1,\dots}) := \sum_{t=0}^{\infty} \gamma^t \mathbb{E}\left[-\log \pi(a_t \mid s_t)\right]$. More background is provided in Appendix D of the supplementary.

**Including preference constraints.** The standard framework can be readily extended to include learner's preferences in the form of constraints on the preference features $\phi_c$. Clearly, the learner's preferences can render exact matching of the teacher's demonstrations infeasible and hence we relax this condition. To this end, we consider the following generic learner model:

$$\max_{\pi,\, \delta_r^{\text{soft}} \geq 0,\, \delta_c^{\text{soft}} \geq 0} \quad H(\pi) - C_r \cdot \|\delta_r^{\text{soft}}\|_p - C_c \cdot \|\delta_c^{\text{soft}}\|_p \tag{1}$$

$$\text{s.t.} \quad |\mu_r(\pi)[i] - \hat{\mu}_r(\Xi^{\mathsf{T}})[i]| \leq \delta_r^{\text{hard}}[i] + \delta_r^{\text{soft}}[i] \ \forall i \in \{1, 2, \dots, d_r\}$$

$$g_j(\mu_c(\pi)) \leq \delta_c^{\text{hard}}[j] + \delta_c^{\text{soft}}[j] \ \forall j \in \{1, 2, \dots, m\},$$

Here, $g\colon \mathbb{R}^{d_c} \mapsto \mathbb{R}$ are $m$ convex functions representing preference constraints. The coefficients $C_r$ and $C_c$ are the learner's parameters which quantify the relative importance of matching the teacher's demonstrations and satisfying the learner's preferences. The learner model is further characterized by parameters $\delta_r^{\mathrm{hard}}[i]$ and $\delta_c^{\mathrm{hard}}[j]$ (we will use the vector notation as $\delta_r^{\mathrm{hard}} \in \mathbb{R}_{\geq 0}^{d_r}$ and $\delta_c^{\mathrm{hard}} \in \mathbb{R}_{\geq 0}^{m}$). The optimization variables for the learner are given by $\pi$, $\delta_r^{\mathrm{soft}}[i]$, and $\delta_c^{\mathrm{soft}}[j]$ (we will use the vector notation as $\delta_r^{\mathrm{soft}} \in \mathbb{R}_{\geq 0}^{d_r}$ and $\delta_c^{\mathrm{soft}} \in \mathbb{R}_{\geq 0}^{m}$). These parameters ($\delta_r^{\mathrm{hard}}$, $\delta_c^{\mathrm{hard}}$) and optimization variables ($\delta_r^{\mathrm{soft}}$, $\delta_c^{\mathrm{soft}}$) characterize the following behavior:

- While a mismatch of up to $\delta_r^{\mathrm{hard}}$ between the learner's and teacher's reward feature expectations incurs no cost regarding the optimization objective, a mismatch larger than $\delta_r^{\mathrm{hard}}$ incurs a cost of $C_r \cdot \|\delta_r^{\mathrm{soft}}\|_p$.
- Similarly, while a violation of up to $\delta_c^{\mathrm{hard}}$ of the learner's preference constraints incurs no cost regarding the optimization objective, a violation larger than $\delta_c^{\mathrm{hard}}$ incurs a cost of $C_c \cdot \|\delta_c^{\mathrm{soft}}\|_p$.

Next, we discuss two special instances of this generic learner model.

## 3.1 Learner Model with Hard Preference Constraints

It is instructive to study a special case of the above-mentioned generic learner model. Let us consider the model in Eq. 1 with $\delta_r^{\mathrm{hard}} = 0$, $\delta_c^{\mathrm{hard}} = 0$, and a limiting case with $C_r, C_c \gg 0$ such that the term $H(\pi)$ can be neglected. Now, if we additionally assume that $C_c \gg C_r$, the learner's objective can be thought of as finding a policy $\pi$ that minimizes the $L^p$ norm distance to the reward feature expectations of the teacher's demonstration while satisfying the constraints $g_j(\mu_c(\pi)) \leq 0 \; \forall j \in \{1, 2, \ldots, m\}$. More formally, we study the following learner model given in Eq. 2 below:

$$\min_{\pi} \quad \|\mu_r(\pi) - \hat{\mu}_r(\Xi^{\mathsf{T}})\|_p \tag{2}$$
$$\text{s.t.} \quad g_j(\mu_c(\pi)) \leq 0 \; \forall j \in \{1, 2, \ldots, m\}.$$

To get a better understanding of the model, we can define the learner's constraint set as $\Omega^{\mathsf{L}} := \{\mu : \mu \in \Omega \text{ s.t. } g_j(\mu_c) \leq 0 \; \forall j \in \{1, 2, \ldots, m\}\}$. Similar to $\Omega^{\mathsf{L}}$, we define $\Omega_r^{\mathsf{L}} \subseteq \Omega_r$ where $\Omega_r^{\mathsf{L}}$ is the projection of the set $\Omega^{\mathsf{L}}$ to the subspaces $\mathbb{R}^{d_r}$. We can now rewrite the above optimization problem as $\min_{\pi \colon \mu_r(\pi) \in \Omega_r^{\mathsf{L}}} \|\mu_r(\pi) - \hat{\mu}_r(\Xi^{\mathsf{T}})\|_p$. Hence, the learner's behavior is given by:

(i) *Learner can match:* When $\hat{\mu}_r(\Xi^{\mathsf{T}}) \in \Omega_r^{\mathsf{L}}$, the learner outputs a policy $\pi^{\mathsf{L}}$ s.t. $\mu_r(\pi^{\mathsf{L}}) = \hat{\mu}_r(\Xi^{\mathsf{T}})$.

(ii) *Learner cannot match:* Otherwise, the learner outputs a policy $\pi^{\mathsf{L}}$ such that $\mu_r(\pi^{\mathsf{L}})$ is given by the $L^p$ norm projection of the vector $\hat{\mu}_r(\Xi^{\mathsf{T}})$ onto the set $\Omega_r^{\mathsf{L}}$.

Figure 1 provides an illustration of the behavior of this learner model. We will design learner-aware teaching algorithms for this learner model in Section 4.1 and Section 5.

## 3.2 Learner Model with Soft Preference Constraints

Another interesting learner model that we study in this paper arises from the generic learner when we consider $m = d_c$ number of box-type linear constraints with $g_j(\mu_c(\pi)) = \mu_c(\pi)[j] \; \forall j \in \{1, 2, \ldots, d_c\}$. We consider an $L^1$ norm penalty on violation, and for simplicity we consider $\delta_r^{\mathrm{hard}}[i] = 0 \; \forall i \in \{1, 2, \ldots, d_r\}$. In this case, the learner's model is given by

$$\max_{\pi, \, \delta_r^{\mathrm{soft}} \geq 0, \, \delta_c^{\mathrm{soft}} \geq 0} \quad H(\pi) - C_r \cdot \|\delta_r^{\mathrm{soft}}\|_1 - C_c \cdot \|\delta_c^{\mathrm{soft}}\|_1 \tag{3}$$
$$\text{s.t.} \quad |\mu_r(\pi)[i] - \hat{\mu}_r(\Xi^{\mathsf{T}})[i]| \leq \delta_r^{\mathrm{soft}}[i] \; \forall i \in \{1, 2, \ldots, d_r\}$$
$$\mu_c(\pi)[j] \leq \delta_c^{\mathrm{hard}}[j] + \delta_c^{\mathrm{soft}}[j] \; \forall j \in \{1, 2, \ldots, d_c\},$$

The solution to the above problem corresponds to a *softmax* policy with a reward function $R_{\boldsymbol{\lambda}}(s) = \langle \boldsymbol{w}_{\boldsymbol{\lambda}}, \phi(s) \rangle$ where $\boldsymbol{w}_{\boldsymbol{\lambda}} \in \mathbb{R}^d$ is parametrized by $\boldsymbol{\lambda}$. The optimal parameters $\boldsymbol{\lambda}$ can be computed efficiently and the corresponding softmax policy is then obtained by *Soft-Value-Iteration* procedure (see [Ziebart, 2010, Algorithm. 9.1], [Zhou et al., 2018]). Details are provided in Appendix E of the supplementary. We will design learner-aware teaching algorithms for this learner model in Section 4.2.

# 4 Learner-aware Teaching under Known Constraints

In this section, we analyze the setting when the teacher has full knowledge of the learner's constraints.

## 4.1 A Learner-aware Teacher for Hard Preferences: AWARE-CMDP

Here, we design a learner-aware teaching algorithm when considering the learner from Section 3.1. Given that the teacher has full knowledge of the learner's preferences, it can compute an optimal teaching policy by maximizing the reward over policies that satisfy the learner's preference constraints, i.e., the teacher solves a constrained-MDP problem (see [De, 1960, Altman, 1999]) given by

$$\max_{\pi} \quad \langle \mathbf{w}_r^*, \mu_r(\pi) \rangle \quad \text{s.t.} \quad \mu_r(\pi) \in \Omega_r^{\mathsf{L}}.$$

We refer to an optimal solution of this problem as $\pi^{\text{aware}}$ and the corresponding teacher as AWARE-CMDP. We can make the following observation formalizing the value of learner-aware teaching:

**Theorem 1.** *For simplicity, assume that the teacher can provide an exact feature expectation $\mu(\pi)$ of a policy instead of providing demonstrations to the learner. Then, the value of learner-aware teaching is*

$$\max_{\pi \text{ s.t. } \mu_r(\pi) \in \Omega_r^{\mathsf{L}}} \left\langle \mathbf{w}_r^*, \mu_r(\pi) \right\rangle - \left\langle \mathbf{w}_r^*, \text{Proj}_{\Omega_r^{\mathsf{L}}} \left( \mu_r(\pi^*) \right) \right\rangle \geq 0.$$

When the set $\Omega^{\mathsf{L}}$ is defined via a set of linear constraints, the above problem can be formulated as a linear program and solved exactly. Details are provided in Appendix F the supplementary material.

## 4.2 A Learner-aware Teacher for Soft Preferences: AWARE-BIL

For the learner models in Section 3, the optimal learner-aware teaching problem can be naturally formalized as the following bi-level optimization problem:

$$\max_{\pi^{\mathsf{T}}} \quad R(\pi^{\mathsf{L}}) \quad \text{s.t.} \quad \pi^{\mathsf{L}} \in \arg\max_{\pi} \text{IRL}(\pi, \mu(\pi^{\mathsf{T}})), \tag{4}$$

where $\text{IRL}(\pi, \mu(\pi^{\mathsf{T}}))$ stands for the IRL problem solved by the learner given demonstrations from $\pi^{\mathsf{T}}$ and can include preferences of the learner (see Eq. 1 in Section 3).

There are many possibilities for solving this bi-level optimization problem—see for example [Sinha et al., 2018] for an overview. In this paper we adopted a *single-level reduction* approach to simplify the above bi-level optimization problem as this results in particularly intuitive optimiziation problems for the teacher. The basic idea of single-level reduction is to replace the lower-level problem, i.e., $\arg\max_{\pi} \text{IRL}(\pi, \mu(\pi^{\mathsf{T}}))$, by the optimality conditions for that problem given by the Karush-Kuhn-Tucker conditions [Boyd and Vandenberghe, 2004, Sinha et al., 2018]. For the learner model outlined in Section 3.2, these reductions take the following form (see Appendix G in the supplementary material for details):

$$\max_{\boldsymbol{\lambda} := \{\boldsymbol{\alpha}^{\text{low}} \in \mathbb{R}^{d_r}, \, \boldsymbol{\alpha}^{\text{up}} \in \mathbb{R}^{d_r}, \, \boldsymbol{\beta} \in \mathbb{R}^{d_c}\}} \quad \langle \mathbf{w}_r^*, \mu_r(\pi_{\boldsymbol{\lambda}}) \rangle \tag{5}$$

$$\text{s.t.} \quad 0 \leq \boldsymbol{\alpha}^{\text{low}} \leq C_r$$
$$0 \leq \boldsymbol{\alpha}^{\text{up}} \leq C_r$$
$$\{0 \leq \boldsymbol{\beta} \leq C_c \ \text{AND} \ \mu_c(\pi_{\boldsymbol{\lambda}}) \leq \delta_c^{\text{hard}}\} \ \text{OR} \ \{\boldsymbol{\beta} = C_c \ \text{AND} \ \mu_c(\pi_{\boldsymbol{\lambda}}) \geq \delta_c^{\text{hard}}\}$$

where $\pi_{\boldsymbol{\lambda}}$ corresponds to a *softmax* policy with a reward function $R_{\boldsymbol{\lambda}}(s) = \langle \boldsymbol{w}_{\boldsymbol{\lambda}}, \phi(s) \rangle$ for $\boldsymbol{w}_{\boldsymbol{\lambda}} = [(\boldsymbol{\alpha}^{\text{low}} - \boldsymbol{\alpha}^{\text{up}})^{\dagger}, -\boldsymbol{\beta}^{\dagger}]^{\dagger}$. Thus, finding optimal demonstrations means optimization over *softmax* teaching policies while respecting the learner's preferences. To actually solve the above optimization problem and find good teaching policies, we use an approach inspired by the Frank-Wolfe algorithm [Jaggi, 2013] detailed in Appendix G of the supplementary material. We refer to a teacher implementing this approach as AWARE-BIL.

# 5 Learner-Aware Teaching Under Unknown Constraints

In this section, we consider the more realistic and challenging setting in which the teacher $\mathsf{T}$ does *not* know the learner $\mathsf{L}$'s constraint set $\Omega_r^{\mathsf{L}}$. Without feedback from $\mathsf{L}$, $\mathsf{T}$ can generally not do better than

the agnostic teacher who simply ignores any constraints. We therefore assume that $\mathsf{T}$ and $\mathsf{L}$ interact in rounds as described by Algorithm 1. The two versions of the algorithm we describe in Sections 5.1 and 5.2 are obtained by specifying how $\mathsf{T}$ adapts the teaching policy in each round.

---

**Algorithm 1** Teacher-learner interaction in the adaptive teaching setting

---

1: Initial teaching policy $\pi^{\mathsf{T},0}$ (e.g., optimal policy ignoring any constraints)
2: **for** round $i = 0, 1, 2, \ldots$ **do**
3:     Teacher provides demonstrations with feature vector $\mu_r^{\mathsf{T},i}$ using policy $\pi^{\mathsf{T},i}$
4:     Learner upon receiving $\mu_r^{\mathsf{T},i}$ computes a policy $\pi^{\mathsf{L},i}$ with feature vector $\mu_r^{\mathsf{L},i}$
5:     Teacher observes learner's feature vector $\mu_r^{\mathsf{L},i}$ and adapts the teaching policy

---

In this section, we assume that $\mathsf{L}$ is as described in Section 3.1: Given demonstrations $\Xi^{\mathsf{T}}$, $\mathsf{L}$ finds a policy $\pi^{\mathsf{L}}$ such that $\mu_r(\pi^{\mathsf{L}})$ matches the $L^2$-projection of $\hat{\mu}_r(\Xi^{\mathsf{T}})$ onto $\Omega_r^{\mathsf{L}}$. For the sake of simplifying the presentation and the analysis, we also assume that $\mathsf{L}$ and $\mathsf{T}$ can observe the exact feature expectations of their respective policies, e.g., $\hat{\mu}_r(\Xi^{\mathsf{T}}) = \mu_r(\pi^{\mathsf{T}})$ if $\Xi^{\mathsf{T}}$ is sampled from $\pi^{\mathsf{T}}$.

### 5.1 An Adaptive Learner-aware Teacher Using Volume Search: ADAWARE-VOL

In our first adaptive teaching algorithm ADAWARE-VOL, $\mathsf{T}$ maintains an estimate $\hat{\Omega}_r^{\mathsf{L}} \supset \Omega_r^{\mathsf{L}}$ of the learner's constraint set, which in each round gets updated by intersecting the current version with a certain affine halfspace, thus reducing the volume of $\hat{\Omega}_r^{\mathsf{L}}$. The new teaching policy is then any policy $\pi^{\mathsf{T},i+1}$ which is optimal under the constraint that $\mu^{\mathsf{T},i+1} \in \hat{\Omega}_r^{\mathsf{L}}$. The interaction ends as soon as $\|\mu_r^{\mathsf{L},i} - \mu_r^{\mathsf{T},i}\|_2 \le \epsilon$ for a threshold $\epsilon$. Details are provided in Appendix C.1 of the supplementary.

**Theorem 2.** *Upon termination of* ADAWARE-VOL*, $\mathsf{L}$'s output policy $\pi^{\mathsf{L}}$ satisfies $R(\pi^{\mathsf{L}}) \ge R(\pi^{\mathrm{aware}}) - \epsilon$ for any policy $\pi^{\mathrm{aware}}$ which is optimal under $\mathsf{L}$'s constraints. For the special case that $\Omega_r^{\mathsf{L}}$ is a polytope defined by $m$ linear inequalities, the algorithm terminates in $O(m^{d_r})$ iterations.*

### 5.2 An Adaptive Learner-aware Teacher Using Line Search: ADAWARE-LIN

In our second adaptive teaching algorithm, ADAWARE-LIN, $\mathsf{T}$ adapts the teaching policy by performing a binary search on a line segment of the form $\{\mu_r^{\mathsf{L},i} + \alpha \mathbf{w}_r^* \mid \alpha \in [\alpha_{\min}, \alpha_{\max}]\} \subset \mathbb{R}^{d_r}$ to find a vector $\mu_r^{\mathsf{T},i+1} = \mu_r^{\mathsf{L},i} + \alpha_i \mathbf{w}_r^*$ that is the vector of feature expectations of a policy; here $\alpha_{\max} > \alpha_{\min} > 0$ are fixed constants. If that is not successful, the teacher finds a teaching policy with $\mu_r^{\mathsf{T},i+1} \in \arg\min_{\mu_r \in \Omega_r} \|\mu_r - \mu_r^{\mathsf{L},i} - \alpha_{\min} \mathbf{w}_r^*\|_2$. The following theorem analyzes the convergence of $\mathsf{L}$'s performance to $\overline{R}_{\mathsf{L}} := \max_{\mu_r \in \Omega_r} R(\mu_r)$ under the assumption that $\mathsf{T}$'s search succeeds in every round. The proof and further details are provided in Appendix C.2 of the supplementary.

**Theorem 3.** *Fix some $\varepsilon > 0$ and assume that there exists a constant $\alpha_{\min} > 0$ such that, as long as $\overline{R}_{\mathsf{L}} - R(\mu_r^{\mathsf{L},i}) > \varepsilon$, the teacher can find a teaching policy $\pi^{\mathsf{T},i+1}$ satisfying $\mu_r^{\mathsf{T},i+1} = \mu_r^{\mathsf{L},i} + \alpha_i \mathbf{w}_r^*$ for some $\alpha_i \ge \alpha_{\min}$. Then the learner's performance increases monotonically in each round of* ADAWARE-LIN*, i.e., $R(\mu_r^{\mathsf{L},i+1}) > R(\mu_r^{\mathsf{L},i})$. Moreover, after at most $O(\frac{D^2}{\varepsilon \alpha_{\min}} \log \frac{D}{\varepsilon})$ teaching steps, the learner's performance satisfies $R(\mu_r^{\mathsf{L},i}) > \overline{R}_{\mathsf{L}} - 2\varepsilon$. Here we abbreviate $D := \operatorname{diam} \Omega_r$.*

## 6 Experimental Evaluation

In this section we evaluate our teaching algorithms for different types of learners on the environment introduced in Figure 1. The environment we consider here has three types of reward objects, i.e., a "star" object with reward of 1.0, a "plus" object with reward of 0.9, and a "dot" object with reward of 0.2. Two objects of each type are placed randomly on the grid such that there is always only a single object in each grid cell. The presence of an object of type "star", "plus", or "dot" in some state $s$ is encoded in the reward features $\phi_r(s)$ by a binary-indicator for each type such that $d_r = 3$. We use a discount factor of $\gamma = 0.99$. Upon collecting an object, there is a 0.1 probability of transiting to a terminal state.

**Learner models.** We consider a total of 5 different learners whose preferences can be described by *distractors* in the environment. Each learner prefers to avoid a certain subset of these distractors.

There is a total of 4 of distractors: (i) two "green" distractors are randomly placed at a distance of 0-cell and 1-cell to the "star" objects, respectively; (ii) two "yellow" distractors are randomly placed at a distance of 1-cell and 2-cells to the "plus" objects, respectively, see Figure 2a.

Through these distractors we define learners L1-L5 as follows: **(L1)** no preference features ($d_c = 0$); **(L2)** two preference features ($d_c = 2$) such that $\phi_c(s)[1]$ and $\phi_c(s)[2]$ are binary indicators of whether there is a "green" distractor at a distance of 0-cells or 1-cell, respectively; **(L3)** four preference features ($d_c = 4$) such that $\phi_c(s)[1]$, $\phi_c(s)[2]$ are as for L2, and $\phi_c(s)[3]$ and $\phi_c(s)[4]$ are binary indicators of whether there is a "green" distractor at a distance of 2-cells or a "yellow" distractor at a distance of 0-cells, respectively; **(L4)** five preference features ($d_c = 5$) such that $\phi_c(s)[1], \ldots, \phi_c(s)[4]$ are as for L3, and $\phi_c(s)[5]$ is a binary indicator whether there is a "yellow" distractor at a distance of 1-cell; and **(L5)** six preference features ($d_c = 6$) such that $\phi_c(s)[1], \ldots, \phi_c(s)[5]$ are as for L4, and $\phi_c(s)[6]$ is a binary indicator whether there is a "yellow" distractor at a distance of 2-cells.

The first row in Figure 2 shows an instance of the considered object-worlds and indicates the preference of the learners to avoid certain regions by the gray area.

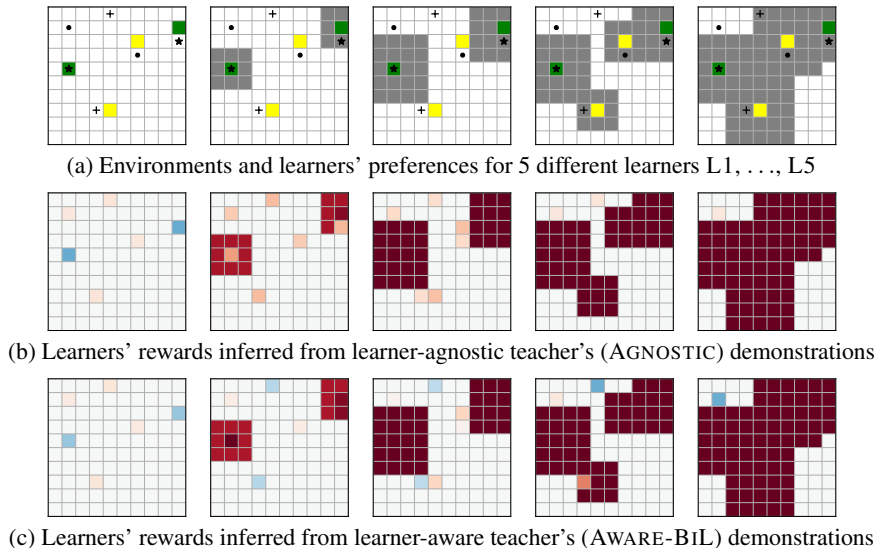

(a) Environments and learners' preferences for 5 different learners L1, ..., L5

(b) Learners' rewards inferred from learner-agnostic teacher's (AGNOSTIC) demonstrations

(c) Learners' rewards inferred from learner-aware teacher's (AWARE-BIL) demonstrations

Figure 2: Teaching in object-world environments under full knowledge of the learner's preferences. Green and yellow cells indicate distractors associated with either "star" or "plus" objects, respectively. Learner's preferences to avoid cells are indicated in gray. Learner model from Section 3.2 with $C_r = 5$, $C_c = 10$, and $\delta_c^{\text{hard}} = 0$ is considered for these experiments. The learner-aware teacher enable the learner to infer reward functions that are compatible with the learner's preferences and achieve higher average rewards. In Figure 2b and Figure 2c, blue color represents positive reward, red color represents negative reward, and the magnitude of the reward is indicated by color intensity.

## 6.1 Teaching under known constraints

In this section we consider learners with soft constraints from Section 3.2, with preference features as described above, and parameters $C_r = 5$, $C_c = 10$, and $\delta_c^{\text{hard}} = 0$ (more experimental results for different values of $C_r$ and $C_c$ are provided in Appendix B.1 of the supplementary). Our first results are presented in Figure 2. The second and third rows show the rewards inferred by the learners for demonstrations provided by a learner-agnostic teacher who ignores any constraints (AGNOSTIC) and the bi-level learner-aware teacher (AWARE-BIL), respectively. We observe that AGNOSTIC fails to teach the learner about objects' positive rewards in cases where the learners' preferences conflict with the position of the most rewarding objects (second row). In contrast, AWARE-BIL always successfully teaches the learners about rewarding objects that are compatible with the learners' preferences (third row).

We also compare AGNOSTIC and AWARE-BIL in terms of reward achieved by the learner after teaching for object worlds of size $10 \times 10$ in Table 1. The numbers show the average reward over 10 randomly generated object-worlds. Note that AWARE-BIL has to solve a non-convex optimization problem to find the optimal teaching policy, cf. Eq. 5. Because we use a gradient-based optimization

approach, the teaching policies found can depend on the initial point for optimization. Hence, we always consider the following two initial points for optimization and select the teaching policy which results in a higher objective value: (i) all optimization variables in Eq. 5 are set to zero, and (ii) the optimization variables are initialized as $\alpha^{\text{low}}[i] = \max\{w_{\boldsymbol{\lambda}}[i], 0\}$, $\alpha^{\text{up}}[i] = \max\{-w_{\boldsymbol{\lambda}}[i], 0\}$, and $\boldsymbol{\beta} = 0$, where $\boldsymbol{w_{\lambda}}$ is as inferred by the learner when taught by AGNOSTIC and $i \in \{1, \ldots, d_r\}$, cf. Section 3.2. From Table 1 we observe that a learner can learn better policies from a teacher that accounts for the learner's preferences.

Table 1: Learners' average rewards after teaching. L1, . . ., L5 correspond to learners with preferences as shown in Figure 2. Results are averaged over 10 random object-worlds, $\pm$ standard error

|  |  | **Learner** ($C_r = 5, C_c = 10$) | | | | |
|---|---|---|---|---|---|---|
|  |  | L1 | L2 | L3 | L4 | L5 |
| **Teacher** | AGNOSTIC | $7.99 \pm 0.02$ | $0.01 \pm 0.00$ | $0.01 \pm 0.00$ | $0.01 \pm 0.00$ | $0.00 \pm 0.00$ |
|  | AWARE-BiL | $8.00 \pm 0.02$ | $7.20 \pm 0.01$ | $4.86 \pm 0.30$ | $3.15 \pm 0.27$ | $1.30 \pm 0.07$ |

## 6.2 Teaching under unknown constraints

In this section we evaluate the teaching algorithms from Section 5. We consider the learner model from Section 3.1 that uses $L^2$-projection to match reward feature expectations as studied in Section 5, cf. Eq. 2.[2] For modeling the hard constraints, we consider box-type linear constraints with $\delta_c^{\text{hard}}[j] = 2.5 \; \forall j \in \{1, 2, \ldots, d_c\}$ for the preference features, cf. Eq. 3.

We study the learners L1, L2, and L3 with preferences corresponding to the first three object-worlds shown in Figure 2a. We report the results for learner L2 below; results for learners L1 and L3 are deferred to the Appendix B.2 of the supplementary material.

In this context it is instructive to investigate how quickly these adaptive teaching strategies converge to the performance of a teacher who has full knowledge about the learner. Results comparing the adaptive teaching strategies (ADAWARE-VOL and ADAWARE-LIN) are shown in Figure 3a. We can observe that both teaching strategies get close to the best possible performance under full knowledge about the learner (AWARE-CMDP). We also provide results showing the performance achieved by the adaptive teaching strategies on object-worlds of varying sizes, see Figure 3b.

Note that the performance of ADAWARE-VOL decreases slightly when teaching for more rounds, i.e., comparing the results after 3 teaching rounds and at the end of the teaching process. This is because of approximations when learner is computing the policy via projection, which in turn leads to errors on the teacher side when approximating $\hat{\Omega}_r^{\mathsf{L}}$ (refer to discussion in Footnote 2). In contrast, ADAWARE-LIN performance always increases when teaching for more rounds.

## 7 Related Work

Our work is closely related to algorithmic machine teaching [Goldman and Kearns, 1995, Zhu, 2015, Zhu et al., 2018], whose general goal is to design teaching algorithms that optimize the data that is provided to a learning algorithm. Most works in machine teaching so far focus on supervised learning tasks and assume that the learning algorithm is fully known to the teacher, see e.g., [Zhu, 2013, Singla et al., 2014, Liu and Zhu, 2016, Mac Aodha et al., 2018].

In the IRL setting, few works study how to provide maximally informative demonstrations to the learner, e.g., [Cakmak and Lopes, 2012, Brown and Niekum, 2019]. In contrast to our work, their teacher fully knows the learner model and provides the demonstrations without any adaptation to the learner. The question of how a teacher should adaptively react to a learner has been addressed by [Singla et al., 2013, Liu et al., 2018, Chen et al., 2018, Melo et al., 2018, Yeo et al., 2019, Hunziker et al., 2019], but only in the supervised setting. In a recent work, [Kamalaruban et al., 2019] have studied the problem of adaptively teaching an IRL agent by pro-

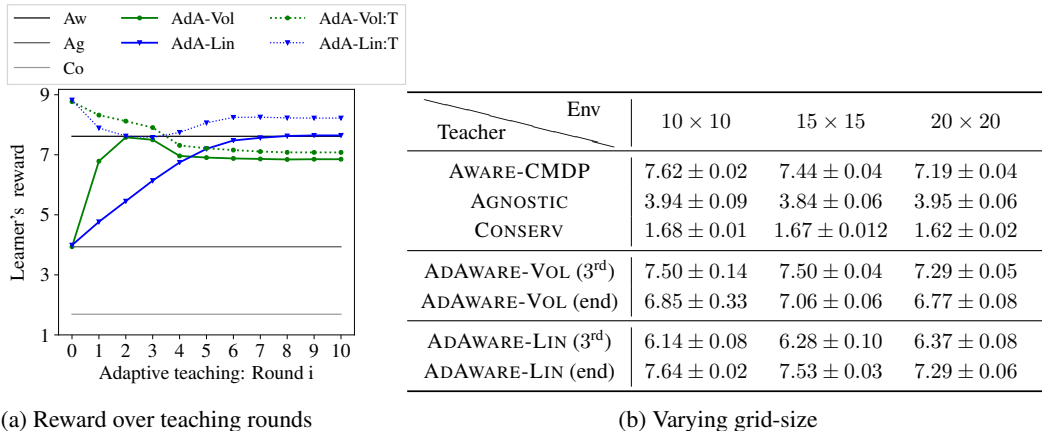

(a) Reward over teaching rounds

| Teacher \ Env | $10 \times 10$ | $15 \times 15$ | $20 \times 20$ |
|---|---|---|---|
| AWARE-CMDP | $7.62 \pm 0.02$ | $7.44 \pm 0.04$ | $7.19 \pm 0.04$ |
| AGNOSTIC | $3.94 \pm 0.09$ | $3.84 \pm 0.06$ | $3.95 \pm 0.06$ |
| CONSERV | $1.68 \pm 0.01$ | $1.67 \pm 0.012$ | $1.62 \pm 0.02$ |
| ADAWARE-VOL ($3^{\text{rd}}$) | $7.50 \pm 0.14$ | $7.50 \pm 0.04$ | $7.29 \pm 0.05$ |
| ADAWARE-VOL (end) | $6.85 \pm 0.33$ | $7.06 \pm 0.06$ | $6.77 \pm 0.08$ |
| ADAWARE-LIN ($3^{\text{rd}}$) | $6.14 \pm 0.08$ | $6.28 \pm 0.10$ | $6.37 \pm 0.08$ |
| ADAWARE-LIN (end) | $7.64 \pm 0.02$ | $7.53 \pm 0.03$ | $7.29 \pm 0.06$ |

(b) Varying grid-size

Figure 3: Performance of adaptive teaching strategies ADAWARE-VOL and ADAWARE-LIN. **(left)** Figure 3a shows the reward for learner's policy over number of teaching interactions. The horizontal lines indicate the performance of learner's policy for the learner-aware teacher with full knowledge of the learner's constraints AWARE-CMDP, the learner-agnostic teacher AGNOSTIC who ignores any constraints, and a conservative teacher CONSERV who considers all 6 constraints (assuming the learner model L5 in Figure 2). Our adaptive teaching strategies ADAWARE-VOL and ADAWARE-LIN significantly outperform baselines (AGNOSTIC and CONSERV) and quickly converge towards the optimal performance of AWARE-CMDP. The dotted lines ADAWARE-VOL:T and ADAWARE-LIN:T show the rewards corresponding to teacher's policy at a round and are shown to highlight the very different behavior of two adaptive teaching strategies. **(right)** Table 3b shows results for varying grid-size of the environment. Results are reported at $i = 3^{\text{rd}}$ round and at the "end" round when algorithm reaches it's stopping criterion. Results are reported as average over 10 runs $\pm$ standard error, where each run corresponds to a random environment.

viding an informative sequence of demonstrations. However, they assume that the teacher has full knowlege of the learner's dynamics.

Within the area of IRL, there is a line of work on active learning approaches [Cohn et al., 2011, Brown et al., 2018, Brown and Niekum, 2018, Amin et al., 2017, Cui and Niekum, 2018], which is related to our work. In contrast to us, they take the perspective of the learner who actively influences the demonstrations it receives. A few papers have addressed the problem that arises when the learner does not have full access to the reward features, e.g., [Levine et al., 2010] and [Haug et al., 2018].

Our work is also loosely related to multi-agent reinforcement learning. [Dimitrakakis et al., 2017] studied the interaction between agents with misaligned models with a focus on the question of how to jointly optimize a policy. [Ghosh et al., 2019] studied the problem of designing robust AI agent that can interact with another agent of unknown type. However, these works do not tackle the problem of teaching an agent by demonstrations. Another related work is [Hadfield-Menell et al., 2016] which studied the cooperation of agents who do not perfectly understand each other.

# 8 Conclusions and Outlook

In this paper we considered inverse reinforcement learning in the context of learners with preferences and constraints. In this setting, the learner does not only focus on matching the teacher's demonstrated behavior but also takes its own preferences, e.g., behavioral biases or physical constraints, into account. We developed a theoretical framework for this setting, and proposed and studied algorithms for learner-aware teaching in which the teacher accounts for the learner's preferences for the cases of known and unknown preference constraints. We demonstrated significant performance improvements of our learner-aware teaching strategies as compared to learner-agnostic teaching both theoretically and empirically. Our theoretical framework and our proposed algorithms foster the application of IRL in real-world settings in which the learner does not blindly follow a teacher's demonstrations.

There are several promising directions for future work, including but not limited to: The evaluation of our approach in machine-human and human-machine tasks; extensions of our approach to other learner models; approaches for learning efficiently from a learner's point of view from a fixed set of (potentially suboptimal) demonstrations in the case of preference constraints.

**Acknowledgements**

This work was supported by Microsoft Research through its PhD Scholarship Programme.

## Footnotes

[2]To implement the learner in Eq. 2, we approximated the learner's projection onto the set $\Omega_r^{\mathsf{L}}$ as follows: We implemented the learner based on the optimization problem given in Eq. 3 with a hard constraint on preferences and $L^2$ norm penalty on reward mismatch scaled with a large value of $C_r = 20$.

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
