[Supplementary Material · neurips19_safe-irl_camera-ready-sup.pdf]

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

# A   List of Appendices

In this section we provide a brief description of the content provided in the appendices of the paper.

- Appendix B provides additional experimental results (Section 6).
- Appendix C provides additional details on the adaptive teaching strategies (Section 5).
- Appendix D provides background on the (discounted) MCE-IRL problem (Section 3).
- Appendix E provides additional details on the (discounted) MCE-IRL problem with preferences (Section 3.2).
- Appendix F provides the LP formulation for the teacher AWARE-CMDP (Section 4.1).
- Appendix G provides additional details on the bi-level optimization approach for the teacher AWARE-BIL (Section 4.2).

# B Experimental Evaluation: Additional Results (Section 6)

## B.1 Teaching under known constraints (Section 6.1)

Additional results for teaching under known constraints are presented in Table 2. We observe that AWARE-BIL clearly outperforms AGNOSTIC for most combinations of $C_r$ and $C_c$. Only for $C_r = 10, C_c = 1$, the teachers AWARE-BIL and AGNOSTIC achieve similar performance because $C_r \gg C_c$, and hence the learner values achieving higher reward more than satisfying its preferences.

Table 2: Learners' average rewards after teaching. L1, ..., L5 correspond to learners with preferences as shown in Figure 2. Results are averaged over 10 random object-worlds, $\pm$ standard error

| | | **Learner** $(C_r = 5, C_c = 10)$ | | | | |
|---|---|---|---|---|---|---|
| | | L1 | L2 | L3 | L4 | L5 |
| **Teacher** | AGNOSTIC | $7.99 \pm 0.02$ | $0.01 \pm 0.00$ | $0.01 \pm 0.00$ | $0.01 \pm 0.00$ | $0.00 \pm 0.00$ |
| | AWARE-BIL | $8.00 \pm 0.02$ | $7.20 \pm 0.01$ | $4.86 \pm 0.30$ | $3.15 \pm 0.27$ | $1.30 \pm 0.07$ |

| | | **Learner** $(C_r = 10, C_c = 10)$ | | | | |
|---|---|---|---|---|---|---|
| | | L1 | L2 | L3 | L4 | L5 |
| **Teacher** | AGNOSTIC | $8.34 \pm 0.01$ | $0.17 \pm 0.02$ | $0.01 \pm 0.00$ | $0.01 \pm 0.00$ | $0.00 \pm 0.00$ |
| | AWARE-BIL | $8.33 \pm 0.01$ | $6.90 \pm 0.17$ | $5.03 \pm 0.31$ | $3.27 \pm 0.28$ | $1.35 \pm 0.07$ |

| | | **Learner** $(C_r = 10, C_c = 5)$ | | | | |
|---|---|---|---|---|---|---|
| | | L1 | L2 | L3 | L4 | L5 |
| **Teacher** | AGNOSTIC | $8.36 \pm 0.01$ | $8.14 \pm 0.03$ | $0.01 \pm 0.00$ | $0.01 \pm 0.00$ | $0.00 \pm 0.00$ |
| | AWARE-BIL | $8.34 \pm 0.01$ | $8.13 \pm 0.03$ | $5.20 \pm 0.29$ | $3.43 \pm 0.27$ | $1.69 \pm 0.0$ |

| | | Learner $(C_r = 5, C_c = 5)$ | | | | |
|---|---|---|---|---|---|---|
| | | L1 | L2 | L3 | L4 | L5 |
| **Teacher** | AGNOSTIC | $7.99 \pm 0.02$ | $0.17 \pm 0.02$ | $0.01 \pm 0.00$ | $0.01 \pm 0.00$ | $0.00 \pm 0.00$ |
| | AWARE-BIL | $8.00 \pm 0.02$ | $6.64 \pm 0.17$ | $4.87 \pm 0.30$ | $3.16 \pm 0.27$ | $1.31 \pm 0.06$ |

| | | **Learner** $(C_r = 10, C_c = 1)$ | | | | |
|---|---|---|---|---|---|---|
| | | L1 | L2 | L3 | L4 | L5 |
| **Teacher** | AGNOSTIC | $8.36 \pm 0.01$ | $8.39 \pm 0.02$ | $8.46 \pm 0.02$ | $8.46 \pm 0.02$ | $8.49 \pm 0.02$ |
| | AWARE-BIL | $8.33 \pm 0.01$ | $8.36 \pm 0.03$ | $8.44 \pm 0.02$ | $8.44 \pm 0.02$ | $8.46 \pm 0.02$ |

| | | **Learner** $(C_r = 1, C_c = 10)$ | | | | |
|---|---|---|---|---|---|---|
| | | L1 | L2 | L3 | L4 | L5 |
| **Teacher** | AGNOSTIC | $5.67 \pm 0.02$ | $0.15 \pm 0.02$ | $0.16 \pm 0.02$ | $0.11 \pm 0.01$ | $0.08 \pm 0.01$ |
| | AWARE-BIL | $5.93 \pm 0.02$ | $4.49 \pm 0.15$ | $3.56 \pm 0.24$ | $2.30 \pm 0.22$ | $0.93 \pm 0.05$ |

## B.2 Teaching under unknown constraints (Section 6.2)

Here, we provide additional experimental results for teaching algorithms from Section 5. In particular, we report on the results for learner L1 and learner L3, similar to the results for learner L2 reported in Section 6.2.

(a) Reward over teaching rounds

| Teacher \ Env | $10 \times 10$ | $15 \times 15$ | $20 \times 20$ |
|---|---|---|---|
| AWARE-CMDP | $8.42 \pm 0.03$ | $8.24 \pm 0.05$ | $7.84 \pm 0.08$ |
| AGNOSTIC | $8.42 \pm 0.03$ | $8.24 \pm 0.05$ | $7.84 \pm 0.08$ |
| CONSERV | $1.68 \pm 0.1$ | $1.66 \pm 0.01$ | $1.65 \pm 0.02$ |
| ADAWARE-VOL ($3^{rd}$) | $8.04 \pm 0.02$ | $7.83 \pm 0.04$ | $7.46 \pm 0.07$ |
| ADAWARE-VOL (end) | $8.06 \pm 0.02$ | $7.80 \pm 0.08$ | $7.30 \pm 0.12$ |
| ADAWARE-LIN ($3^{rd}$) | $8.44 \pm 0.04$ | $8.23 \pm 0.07$ | $8.08 \pm 0.08$ |
| ADAWARE-LIN (end) | $8.44 \pm 0.04$ | $8.23 \pm 0.07$ | $8.08 \pm 0.08$ |

(b) Varying grid-size

Figure 4: Results for learner L1

(a) Reward over teaching rounds

| Teacher \ Env | $10 \times 10$ | $15 \times 15$ | $20 \times 20$ |
|---|---|---|---|
| AWARE-CMDP | $7.68 \pm 0.04$ | $7.35 \pm 0.03$ | $7.39 \pm 0.09$ |
| AGNOSTIC | $3.11 \pm 0.08$ | $3.12 \pm 0.07$ | $3.26 \pm 0.14$ |
| CONSERV | $1.68 \pm 0.01$ | $1.65 \pm 0.01$ | $1.62 \pm 0.01$ |
| ADAWARE-VOL ($3^{rd}$) | $6.16 \pm 0.42$ | $5.72 \pm 0.54$ | $6.39 \pm 0.32$ |
| ADAWARE-VOL (end) | $5.99 \pm 0.46$ | $5.38 \pm 0.56$ | $6.16 \pm 0.31$ |
| ADAWARE-LIN ($3^{rd}$) | $6.25 \pm 0.20$ | $5.13 \pm 0.50$ | $6.15 \pm 0.11$ |
| ADAWARE-LIN (end) | $7.22 \pm 0.16$ | $5.83 \pm 0.62$ | $7.09 \pm 0.07$ |

(b) Varying grid-size

Figure 5: Results for learner L3

# C  Details for Learner-Aware Teaching under Unknown Constraints (Section 5)

In this appendix, we provide more details on the adaptive teaching algorithms ADAWARE-VOL and ADAWARE-LIN described in Sections 5.1 and 5.2. Recall that both teaching algorithms are obtained from Algorithm 1 by defining the way in which the teacher T adapts the teaching policy based on the learner L's feature expectations $\mu_r^\mathsf{L}$ in past rounds.

## C.1  Details for ADAWARE-VOL (Section 5.1)

**Estimation of the learner's constraint set.**  In ADAWARE-VOL, T maintains an estimate $\hat{\Omega}_r^{\mathsf{L},i}$ of L's constraint set, starting with $\hat{\Omega}_r^{\mathsf{L},0} = \Omega_r$. After observing the feature expectations $\mu_r^{\mathsf{L},i}$ of the policy L found in round $i$, T updates this estimate as follows:

$$\hat{\Omega}_r^{\mathsf{L},i+1} := \hat{\Omega}_r^{\mathsf{L},i} \cap \{\mu_r^{\mathsf{L},i} + \nu \in \mathbb{R}^{d_r} \mid \langle \mu_r^{\mathsf{T},i} - \mu_r^{\mathsf{L},i}, \nu \rangle \leq 0\} \tag{6}$$

The set on the right hand side of (6) with which $\Omega_r^{\mathsf{L},i}$ gets intersected is a halfspace containing $\Omega_r^\mathsf{L}$. This is due to the fact that $\Omega_r^\mathsf{L}$ is convex by assumption, and to our assumption that L's learning algorithm is such that it outputs a policy whose feature expectations $\mu_r^{\mathsf{L},i}$ match the $L^2$-projection of $\mu_r^{\mathsf{T},i}$ to $\Omega_r^\mathsf{L}$. Inductively, it follows that $\hat{\Omega}_r^{\mathsf{L},i} \supset \Omega_r^\mathsf{L}$ for all $i$.

In practice, we implement a slightly modified version of the update step in which we intersect $\hat{\Omega}_r^{\mathsf{L},i}$ with a halfspace that is shifted in the direction of $\mu_r^{\mathsf{T},i} - \mu_r^{\mathsf{L},i}$ by a small amount, i.e., we use

$$\{\mu_r^{\mathsf{L},i} + (1-\eta)(\mu_r^{\mathsf{T},i} - \mu_r^{\mathsf{L},i}) + \nu \in \mathbb{R}^{d_r} \mid \langle \mu_r^{\mathsf{T},i} - \mu_r^{\mathsf{L},i}, \nu \rangle \leq 0\}$$

with a step size parameter $\eta \in (0,1)$. This helps make the algorithm more robust to noise in the learner's feature expectations. In our experiments, we used $\eta = 0.9$.

**Update of the teaching policy.**  After updating the estimate of the learner's constraint set to $\hat{\Omega}_r^{\mathsf{L},i}$, T solves a constrained MDP in order to find

$$\pi^{\mathsf{T},i+1} \in \underset{\pi, \mu_r(\pi) \in \hat{\Omega}_r^{\mathsf{L},i}}{\arg\max} \; R(\pi).$$

Given that $\hat{\Omega}_r^{\mathsf{L},i}$ is cut out by linear equations, solving the constrained MDP reduces to solving an LP, as described in Appendix F.

**Termination of the interaction.**  The algorithm terminates as soon as the stopping criterion $\|\mu_r^{\mathsf{L},i} - \mu_r^{\mathsf{T},i}\|_2 \leq \epsilon$ is satisfied. Note that $\hat{\Omega}_r^{\mathsf{L},i} \supset \Omega_r^\mathsf{L}$ implies that

$$R(\pi^{\mathsf{T},i}) \geq R(\pi^{\text{aware}})$$

for any $\pi^{\text{aware}} \in \arg\max_{\pi, \mu_r(\pi) \in \Omega_r^\mathsf{L}} R(\pi)$. Therefore, after termination we have

$$R(\pi^{\mathsf{L},i}) \geq R(\pi^{\text{aware}}) - \epsilon$$

for any policy $\pi^{\text{aware}}$ which is optimal under L's constraints, which is the first statement of Theorem 2.

The second statement of Theorem 2 follows from the fact that if $\Omega_r^\mathsf{L}$ is a convex polytope cut out by $m$ linear inequalities, the number of faces, which is in $O(m^{d_r})$, is an upper bound on the number of iterations of the algorithm, because one face is "eliminated" in each round.

## C.2  Details for ADAWARE-LIN (Section 5.2)

In ADAWARE-LIN, T updates the teaching policy $\pi^{\mathsf{T},i+1}$ based on L's feature expectations $\mu_r^{\mathsf{L},i}$ from the previous round. To do so, T uses LINESEARCH (Algorithm 2) to perform a binary search on the line segment

$$\{\mu_r^{\mathsf{L},i} + \alpha \mathbf{w}_r^* \mid \alpha \in [\alpha_{\min}, \alpha_{\max}]\} \subset \mathbb{R}^{d_r} \tag{7}$$

in order to find a vector $\mu_r$ that is realizable as the vector of feature expectations of a policy. If the intersection of the line segment (7) with $\Omega_r$ is non-empty, it is of the form $\{\mu_r^\mathsf{L} + \alpha \mathbf{w}_r^* \mid \alpha \in$

**Algorithm 2** LINESEARCH

**Require:** $\mu_r^{\mathsf{L}}, \alpha_{\min}, \alpha_{\max}, \varepsilon_\alpha, \varepsilon_\mu$.
1: $\alpha_u \leftarrow \alpha_{\max}, \alpha_l \leftarrow \alpha_{\min}$
2: **while** $\alpha_u - \alpha_l > \varepsilon_\alpha$ **do**
3:     $\alpha \leftarrow (\alpha_u + \alpha_l)/2$
4:     $\pi^{\mathsf{T}} \leftarrow \mathrm{IRL}(\mu_r^{\mathsf{L}} + \alpha \mathbf{w}_r^*)$
5:     **if** $\|\mu_r(\pi^{\mathsf{T}}) - \mu_r^{\mathsf{L}} - \alpha \mathbf{w}_r^*\|_2 > \varepsilon_\mu$ **then**
6:         $\alpha_u \leftarrow \alpha$
7:     **else**
8:         $\alpha_l \leftarrow \alpha$
9: **if** $\|\mu_r(\pi^{\mathsf{T}}) - \mu_r^{\mathsf{L}} - \alpha \mathbf{w}_r^*\|_2 > \varepsilon_\mu$ **then**
10:     $\pi^{\mathsf{T}} \leftarrow \mathrm{IRL}(\mu_r^{\mathsf{L}} + \alpha_{\min} \mathbf{w}_r^*)$
11: **return** $\pi^{\mathsf{T}}$

Figure 6

LINESEARCH is the algorithm that $\mathsf{T}$ uses in order to find a teaching policy $\pi^{\mathsf{T}}$ provided that the feature expectations of L's current policy are $\mu_r^{\mathsf{L}}$. Figure 6 illustrates the two cases may occur: For the right $\mu_r^{\mathsf{L}}$, LINESEARCH returns a policy $\pi^{\mathsf{T}}$ whose feature expectations satisfy $\mu_r^{\mathsf{T}} = \mu_r^{\mathsf{L}} + \alpha^* \mathbf{w}_r^*$ such that $\alpha^* > \alpha_{\min}$. For the left $\mu_r^{\mathsf{L}}$, LINESEARCH returns a policy $\pi^{\mathsf{T}}$ whose feature expectations satisfy $\mu_r^{\mathsf{T}} \in \arg\min_{\mu_r \in \Omega_r} \|\mu_r - \mu_r^{\mathsf{L}} + \alpha_{\min} \mu_r^{\mathsf{T}}\|$.

$[\alpha_{\min}, \alpha^*]\}$ for some $\alpha^* \leq \alpha_{\max}$ due to the convexity of $\Omega_r$. In that case, LINESEARCH returns a policy with feature expectations

$$\mu_r^{\mathsf{T},i+1} = \mu_r^{\mathsf{L},i} + \alpha_i^* \mathbf{w}_r^*,$$

where $\alpha_i^*$ is the maximal $\alpha \in [\alpha_{\min}, \alpha_{\max}]$ such that $\mu_r^{\mathsf{L},i} + \alpha \mathbf{w}_r^* \in \Omega_r$. If the intersection is empty, LINESEARCH returns a policy with feature expectations

$$\mu_r^{\mathsf{T},i+1} \in \arg\min_{\mu_r \in \Omega_r} \|\mu_r - \mu_r^{\mathsf{L},i} - \alpha_{\min} \mathbf{w}_r^*\|_2.$$

Figure 6 illustrates the two cases that may occur.

### C.2.1 Proof of Theorem 3

In this section, we provide the proof of Theorem 3, which gives a guarantee on the improvement of L's performance in each round of the ADAWARE-LIN algorithm. The assumption we make here is that, in every teaching round, LINESEARCH returns a teaching policy $\pi^{\mathsf{T},i+1}$ such that $\mu_r^{\mathsf{T},i+1} = \mu_r^{\mathsf{L},i} + \alpha_i \mathbf{w}_r^*$ for some $\alpha_i \geq \alpha_{\min}$, where $\alpha_{\min} > 0$ is a fixed constant. It is easy to see that this assumption, together with our assumption on L's algorithm and the convexity of $\Omega_r^{\mathsf{L}}$, imply that the change in learner performance

$$\Delta R_i := R(\mu_r^{\mathsf{L},i+1}) - R(\mu_r^{\mathsf{L},i})$$

is non-negative in every teaching round. The following proposition, which will be needed in the proof of Theorem 3, strengthens this statement:

**Proposition 1.** *Let* $\overline{R}_{\mathsf{L}} := \max_{\mu_r \in \Omega_r^{\mathsf{L}}} R(\mu_r)$ *be the maximally achievable learner performance. Assume that, in teaching round* $i$, $\mathsf{T}$ *can find a teaching policy* $\pi^{\mathsf{T},i+1}$ *whose feature expectations satisfy* $\mu_r^{\mathsf{T},i+1} = \mu_r^{\mathsf{L},i} + \alpha_i \mathbf{w}_r^*$ *for some* $\alpha_i > 0$. *Then*

$$\overline{R}_{\mathsf{L}} - R(\mu_r^{\mathsf{L},i}) \leq \Delta R_i + D \cdot \sqrt{\frac{\Delta R_i}{\alpha_i - \Delta R_i}}, \tag{8}$$

*where* $D = \operatorname{diam}\Omega_r$.

*Proof of Proposition 1.* Consider the plane $V \subset \mathbb{R}^{d_r}$ spanned by $\mu_r^{\mathsf{L},i}, \mu_r^{\mathsf{T},i+1}$ and $\mu_r^{\mathsf{L},i+1}$ and denote by $\tilde{\mu}_r$ the unique point in $V$ with the properties that

(a) $\langle \mathbf{w}_r^*, \tilde{\mu}_r \rangle = \langle \mathbf{w}_r^*, \mu_r^{\mathsf{L},i+1} \rangle$,

(b) $\tilde{\mu}_r$ lies on the same side of the line through $\mu^{\mathsf{L},i}$ and $\mu^{\mathsf{T},i+1}$ as $\mu_r^{\mathsf{L},i+1}$, and

(c) $\tilde{\mu}_r, \mu_r^{\mathsf{T},i+1}$ and $\mu_r^{\mathsf{L},i}$ span a right triangle with $\tilde{\mu}_r$ at the right-angled corner.

Note that $\mu_r^{\mathsf{L},i+1}$ must lie inside this triangle, i.e., on the red line segment in Figure 7: Otherwise there would a point on the line segment connecting $\mu_r^{\mathsf{L},i+1}$ and $\mu_r^{\mathsf{L},i}$, and hence in $\Omega_r^{\mathsf{L}}$ by convexity, which is closer to $\mu_r^{\mathsf{T},i+1}$ than $\mu_r^{\mathsf{L},i+1}$, contradicting the fact that $\mu_r^{\mathsf{L},i+1}$ is closest to $\mu_r^{\mathsf{T},i+1}$ among all points in $\Omega_r^{\mathsf{L}}$. Denote by $\tilde{\ell}$ the line passing through $\tilde{\mu}_r$ and $\mu_r^{\mathsf{L},i}$.

Figure 7: Illustration of the proof of Proposition 1: The smaller the performance increase $\Delta R_i$, the better the upper bound on the gap $\overline{R}_\Omega - R(\mu_r^{\mathsf{L},i})$.

The facts that $\Omega_r^{\mathsf{L}}$ is convex and that $\mu_r^{\mathsf{L},i+1} = \arg\min_{\mu_r \in \Omega_r^{\mathsf{L}}} \|\mu_r^{\mathsf{T},i+1} - \mu_r\|_2$ imply that $\Omega_r^{\mathsf{L}}$ must lie on one side of the hyperplane

$$\mu_r^{\mathsf{L},i+1} + (\mu_r^{\mathsf{T},i+1} - \mu_r^{\mathsf{L},i+1})^\perp \subset \mathbb{R}^{d_r}.$$

Therefore, we can upper bound $\overline{R}_{\mathsf{L}}$ in terms of the slope $s_\ell$ of the line $\ell$ which arises by intersecting that hyperplane with $V$:

$$\overline{R}_{\mathsf{L}} \le R(\mu_r^{\mathsf{L},i+1}) + D \cdot s_\ell = R(\mu_r^{\mathsf{L},i}) + \Delta R_i + D \cdot s_\ell. \tag{9}$$

Note that the slope $s_\ell$ is upper bounded by the slope $s_{\tilde{\ell}}$ of $\tilde{\ell}$. We have $s_{\tilde{\ell}} = \frac{\Delta R_i}{h}$, where $h$ is the length of the red line segment in Figure 7, and $h = \sqrt{(\alpha_i - \Delta R_i)\Delta R_i}$ by Pythagoras's theorem. Using that, we obtain

$$s_\ell \le s_{\tilde{\ell}} = \sqrt{\frac{\Delta R_i}{\alpha_i - \Delta R_i}}. \tag{10}$$

The claimed estimate (8) follows by plugging this upper bound for $s$ into (9) and rearranging. $\qquad\square$

**Proof of Theorem 3.**

*Proof of Theorem 3.* The fact that $R(\mu_r^{\mathsf{L},i+1}) > R(\mu_r^{\mathsf{L},i})$, which is equivalent to $\Delta R_i > 0$, follows immediately from Proposition 1.

We now prove the claimed rate of convergence.

First, using Proposition 1, we note that the assumption that $\overline{R}_{\mathsf{L}} - R(\mu_r^{\mathsf{L},i}) > \varepsilon$ implies that

$$\varepsilon < \Delta R_i + D\sqrt{\frac{\Delta R_i}{\alpha_i - \Delta R_i}}. \tag{11}$$

Using that, we can conclude that

$$\sqrt{\Delta R_i} > \min\{\sqrt{\varepsilon/2}, \varepsilon\sqrt{\alpha_{\min}/(4D^2 + \varepsilon^2)}\}. \tag{12}$$

Indeed, if $\Delta R_i \le \frac{\varepsilon}{2}$, it follows from (11) that we must have $D \cdot \sqrt{\Delta R_i/(\alpha_{\min} - \Delta R_i)} > \frac{\varepsilon}{2}$, which implies $\sqrt{\Delta R_i} > \varepsilon\sqrt{\alpha_{\min}/(4D^2 + \varepsilon^2)}$. Since we are interested in the behavior as $\varepsilon \to 0$, we assume from now on that $\varepsilon$ is so small that $\varepsilon\sqrt{\alpha_{\min}/(4D^2 + \varepsilon^2)} < \sqrt{\varepsilon/2}$, so that (12) becomes

$$\sqrt{\Delta R_i} > \varepsilon\sqrt{\alpha_{\min}/(4D^2 + \varepsilon^2)} =: C_0. \tag{13}$$

Second, we observe that

$$\sqrt{\alpha_i - \Delta R_i} > \sqrt{\frac{\alpha_{\min}}{2}} =: C_1 \tag{14}$$

except in at most $N := \frac{2}{\alpha_{\min}}(\max R|_\Omega - \min R|_\Omega)$ teaching steps. To see that, note that if the claimed inequality, which is equivalent to $\alpha_i - \frac{\alpha_{\min}}{2} > \Delta R_i$, does not hold, performance increases by at least $\Delta R_i \geq \frac{\alpha_{\min}}{2}$ as $\alpha_i > \alpha_{\min}$, and that can happen at most $N$ times.

The inequalities (13) and (14) together imply that we have

$$C_0 \cdot C_1 \leq \sqrt{(\alpha_i - \Delta R_i)\Delta R_i} \tag{15}$$

as long as $\overline{R}_\mathsf{L} - R(\mu_r^{\mathsf{L},i}) > \varepsilon$, except in at most $N$ teaching steps. Setting $C := \frac{1}{C_0 \cdot C_1}$, this is equivalent to

$$\sqrt{\frac{\Delta R_i}{\alpha_i - \Delta R_i}} \leq C\Delta R_i \tag{16}$$

Plugging (16) into the bound (8) provided by Proposition 1, we obtain the estimate

$$\frac{1}{1 + CD}(\overline{R}_\mathsf{L} - R(\mu_r^{\mathsf{L},i})) \leq \Delta R_i. \tag{17}$$

We have $C = \frac{1}{\varepsilon\alpha_{\min}}\sqrt{2(4D^2 + \varepsilon^2)}$, and hence

$$\frac{1}{1 + CD} = \frac{\varepsilon\alpha_{\min}}{\varepsilon\alpha_{\min} + \sqrt{2(4D^2 + \varepsilon^2)} \cdot D} \geq \frac{1}{1 + \sqrt{10}}\frac{\varepsilon\alpha_{\min}}{D^2} =: \lambda \tag{18}$$

If we had the estimates (17), (18) for *all* teaching steps, we could conclude that the learner performance satisfies $R(\mu_r^{\mathsf{L},i}) > \overline{R}_\mathsf{L} - 2\varepsilon$ after at most $O(\frac{D^2}{\varepsilon\alpha_{\min}}\log\frac{D}{\varepsilon})$ teaching steps. One can see that e.g. by comparing the sequence $R_0, R_1, R_2, \ldots$ with the solution $R(t)$ of the ordinary differential equation $\dot{R} = \lambda(\overline{R}_\mathsf{L} - R)$, which satifies $\overline{R}_\mathsf{L} - R(t) = (\overline{R}_\mathsf{L} - R(0))\exp(-\lambda t)$. Since the number $N$ of teaching steps for which (17), (18) do potentially *not* hold is $O(\frac{D}{\alpha_{\min}})$, we can still make this conclusion. $\square$

## D   Background on (discounted) MCE-IRL Problem (Section 3)

Our learner models build on the (discounted) Maximum Causal Entropy (MCE) IRL framework [Ziebart et al., 2008, Ziebart, 2010, Ziebart et al., 2013, Zhou et al., 2018]. The results below are based on the MDCE-IRL formulation from [Zhou et al., 2018].

### D.1   Primal problem

In the standard (discounted) MCE-IRL framework, a learning agent aims to identify a policy that matches the feature expectations of the teacher's demonstrations while simultaneously maximizing the (discounted) causal entropy of the policy, i.e., the learner solves the following optimization problem:

$$\max_\pi \quad H^\gamma(A_{0:\infty}\|S_{0:\infty}) := \sum_{t=0}^\infty \gamma^t \mathbb{E}\Big[-\log\pi(a_t \mid s_t)\Big]$$

$$\text{subject to} \quad \mu_r(\pi)[i] = \hat{\mu}_r(\Xi^\mathsf{T})[i] \quad \forall i \in \{1, 2, \ldots, d_r\}.$$

Here, $\mu_r(\pi)[i]$ and $\hat{\mu}_r(\Xi^\mathsf{T})[i]$ denote the scalar values of the $i^{\text{th}}$ reward feature. The idea is that without any further information beyond the teacher's demonstrations, the most uncertain solution matching the reward feature expectation of those demonstrations should be preferred.

Formulating this as a minimization problem and spelling out all the constraints, we arrive at the following primal:

$$\min_{\boldsymbol{\pi}=\{\pi_t\}_{t=0}^\infty} -H^\gamma(A_{0:\infty}\|S_{0:\infty})$$

$$\text{subject to}$$

$$\mu_r(\pi_t)[i] = \hat{\mu}_r(\Xi^{\mathsf{T}})[i] \quad \forall i \in \{1, 2, \dots, d_r\}$$
$$\pi_t(a|s) \geq 0 \quad \forall a \in \mathcal{A}, s \in \mathcal{S}, t \geq 0$$
$$\sum_{a \in \mathcal{A}} \pi_t(a|s) = 1 \quad \forall s \in \mathcal{S}, t \geq 0$$
$$\pi_t(a|s) = \pi_{t'}(a|s) \quad \forall a \in \mathcal{A}, s \in \mathcal{S}, t \geq 0, t' \geq 0$$

The last condition ensures that the policy $\pi$ is stationary.

## D.2 Lagrangian relaxation

The Lagrangian relaxation optimization formulation of the above primal problem is given by

$$\mathcal{L}(\boldsymbol{\pi}, \boldsymbol{\lambda}, \boldsymbol{\psi}) = -H^\gamma(A_{0:\infty} \| S_{0:\infty}) + \boldsymbol{\lambda}^\dagger(\hat{\mu}_r(\Xi^{\mathsf{T}}) - \mu_r(\pi_t)) + \sum_{s,t} \psi_{s,t}(1 - \sum_{a \in \mathcal{A}} \pi_t(a|s))$$

subject to
$$\pi_t(a|s) \geq 0 \quad \forall a \in \mathcal{A}, s \in \mathcal{S}, t \geq 0$$
$$\pi_t(a|s) = \pi_{t'}(a|s) \quad \forall a \in \mathcal{A}, s \in \mathcal{S}, t, t' \geq 0$$

Here, $\boldsymbol{\lambda} \in \mathbb{R}^{d_r}$ and $\boldsymbol{\psi} = \{\psi_{s,t}\}_{\forall s_t}$. Also, $\dagger$ is the transpose operator defined for vectors.

*Remark.* The Lagrangian relaxation of the optimization problem is not convex in the problem variables because of the term $\boldsymbol{\lambda}^\dagger(\hat{\mu}_r(\Xi^{\mathsf{T}}) - \mu_r(\pi_t))$ in the objective function, which is not convex in the variables $\pi_t$. However, it can be shown that strong duality holds for both its dual and primal formulations ([Zhou et al., 2018]). The dual formulation is described in Section D.4.

## D.3 Parametric form of the policy

For a given $\boldsymbol{\lambda}$, the optimal policy $\pi_{\boldsymbol{\lambda}}^{\mathrm{soft}}(a|s)$ is given by

$$\pi_{\boldsymbol{\lambda}}^{\mathrm{soft}}(a|s) = \frac{\exp(Q_{\boldsymbol{\lambda}}^{\mathrm{soft}}(s, a))}{\exp(V_{\boldsymbol{\lambda}}^{\mathrm{soft}}(s))}$$

where the quantities are defined recursively as follows:

$$Q_{\boldsymbol{\lambda}}^{\mathrm{soft}}(s, a) = \boldsymbol{\lambda}^\dagger \mu_r(\pi_{\boldsymbol{\lambda}}^{\mathrm{soft}}(a|s)) + \gamma \sum_{s' \in \mathcal{S}} T(s'|s, a) V_{\boldsymbol{\lambda}}^{\mathrm{soft}}(s')$$

$$V_{\boldsymbol{\lambda}}^{\mathrm{soft}}(s) = \log \sum_{a \in \mathcal{A}} \exp(Q_{\boldsymbol{\lambda}}^{\mathrm{soft}}(s, a))$$

This is shown by taking the derivative of the Lagrangian, $\mathcal{L}(\boldsymbol{\pi}, \boldsymbol{\lambda}, \boldsymbol{\psi})$ w.r.t. the primal variables $\pi_t$ and equating it to 0, i.e.,

$$\frac{\partial L(\{\pi_t\}_{t=0}^\infty, \boldsymbol{\lambda}, \boldsymbol{\psi})}{\partial \pi_t} = 0.$$

For a given $\boldsymbol{\lambda}$, the corresponding softmax policy can be obtained by *Soft-Value-Iteration* procedure (see [Ziebart, 2010, Algorithm. 9.1], [Zhou et al., 2018]).

## D.4 Dual problem

For any given $\boldsymbol{\lambda}, \boldsymbol{\psi}$, let $g(\boldsymbol{\lambda}, \boldsymbol{\psi})$ be the optimal value for the optimization problem defined by the Lagrangian relaxation problem in Section D.2. As *strong duality* holds for the (discounted) MCE-IRL problem and its dual counter part, we solve only the following concave dual problem:

$$\underset{\boldsymbol{\lambda} \in \mathbb{R}^{d_r}, \psi_{s,t} \in \mathbb{R}}{\mathrm{maximize}} \quad g(\boldsymbol{\lambda}, \boldsymbol{\psi})$$

## D.5 Gradients for the dual variables

As the dual problem is concave, it can be solved using gradient ascent. The gradients of the dual function described in Section D.4 are given by:

$$
\begin{aligned}
\nabla_{\boldsymbol{\lambda}} \quad & g = \hat{\mu}_r(\Xi^{\mathsf{T}}) - \mu_r(\pi_{\boldsymbol{\lambda}}^{\text{soft}}) \\
\nabla_{\psi_{s,t}} \quad & g = 1 - \sum_{a \in \mathcal{A}} \pi_{\boldsymbol{\lambda}}^{\text{soft}}(a|s)
\end{aligned}
$$

Here $\pi_{\boldsymbol{\lambda}}^{\text{soft}}$ is the parametric softmax policy described above. The second condition is automatically satisfied because $\pi_{\boldsymbol{\lambda}}^{\text{soft}}$ is a probability distribution.

The gradient update rule to compute the optimal $\boldsymbol{\lambda}$ is:

$$
\boldsymbol{\lambda}_{\text{next}} \leftarrow \boldsymbol{\lambda} - \eta \cdot \left( \mu_r(\pi_{\boldsymbol{\lambda}}^{\text{soft}}) - \hat{\mu}_r(\Xi^{\mathsf{T}}) \right)
$$

where $\eta$ is the learning rate.

# E Details of (discounted) MCE-IRL Problem with Preferences (Section 3.2)

Here we present the background of the learner model described in Section 3.2. In this setting, the learner's preferences are modeled as linear soft constraints with L1 penalties. We consider the minimization variant of the problem. The results in this section follow directly from the analysis of Maximum Entropy Models under different constraints, as presented in [Kazama and Tsujii, 2005, Dudík et al., 2007] when applied to (discounted) MCE-IRL problem [Ziebart et al., 2013, Zhou et al., 2018]. For brevity, redundant details of the derivations are omitted. The final policy of the learner is given by $\pi_{\boldsymbol{\lambda}}^{\text{soft}}$ and is defined in Section E.3.

## E.1 Primal problem

The primal problem is given by

$$
\min_{\boldsymbol{\pi} = \{\pi_t\}_{t=0}^{\infty};\, \delta_r^{\text{soft,low}},\, \delta_r^{\text{soft,up}},\, \delta_c^{\text{soft,up}} \geq 0} - H^{\gamma}(A_{0:\infty} || S_{0:\infty}) + \sum_{i=1}^{d_r} C_r \cdot (\delta_r^{\text{soft,low}}[i] + \delta_r^{\text{soft,up}}[i]) + \sum_{j=1}^{d_c} C_c \cdot \delta_c^{\text{soft,up}}[j]
$$

subject to

$$
\begin{aligned}
\hat{\mu}_r(\Xi^{\mathsf{T}})[i] - \mu_r(\pi_t)[i] &\leq \delta_r^{\text{soft,low}}[i] \quad \forall i \in \{1, 2, \ldots, d_r\} \\
\mu_r(\pi_t)[i] - \hat{\mu}_r(\Xi^{\mathsf{T}})[i] &\leq \delta_r^{\text{soft,up}}[i] \quad \forall i \in \{1, 2, \ldots, d_r\} \\
\mu_c(\pi_t)[j] &\leq \delta_c^{\text{hard}}[j] + \delta_c^{\text{soft,up}}[j] \quad \forall j \in \{1, 2, \ldots, d_c\}
\end{aligned}
$$

Here we have $\delta_r^{\text{soft,low}}, \delta_r^{\text{soft,up}} \in \mathbb{R}^{d_r}$ and $\delta_c^{\text{soft,up}} \in \mathbb{R}^{d_c}$ as the primal optimization slack variables with the constraint that $\delta_r^{\text{soft,low}}, \delta_r^{\text{soft,up}}, \delta_c^{\text{soft,up}} \geq 0$. We also have $C_r > 0, C_c > 0$. $\delta_c^{\text{hard}} \in \mathbb{R}^{d_c}$ is a given constant vector.

*Remark.* low and up in the superscripts of dual variables represent whether they are variables for lower bound constraints or upper bound constraints.

## E.2 Lagrangian relaxation

The Lagrangian relaxation optimization formulation of the primal problem described in Section E.1 is given by

$$
\begin{aligned}
\mathcal{L}(\boldsymbol{\pi}, \delta_r^{\text{soft,low}}, \delta_r^{\text{soft,up}}, \delta_c^{\text{soft,up}}, \boldsymbol{\lambda}, \boldsymbol{\psi}) = & -H^{\gamma}(A_{0:\infty}, S_{0:\infty}) + (\boldsymbol{\alpha}^{\text{low}} - \boldsymbol{\alpha}^{\text{up}})^{\dagger}(\hat{\mu}_r(\Xi^{\mathsf{T}}) - \mu_r(\pi_t)) \\
& + \boldsymbol{\beta}^{\dagger} \mu_c(\pi_t) \\
& + \sum_{s,t} \psi_{s,t}(1 - \sum_{a \in \mathcal{A}} \pi_t(a|s)) - (\boldsymbol{\alpha}^{\text{low}})^{\dagger} \delta_r^{\text{soft,low}} - (\boldsymbol{\alpha}^{\text{up}})^{\dagger} \delta_r^{\text{soft,up}} \\
& - \boldsymbol{\beta}^{\dagger} \delta_c^{\text{soft,up}} - \boldsymbol{\beta}^{\dagger} \delta_c^{\text{hard}}
\end{aligned}
$$

$$- (\boldsymbol{\rho}^{\text{low}})^{\dagger} \delta_r^{\text{soft,low}} - (\boldsymbol{\rho}^{\text{up}})^{\dagger} \delta_r^{\text{soft,up}}$$
$$- \boldsymbol{\sigma}^{\dagger} \delta_c^{\text{soft,up}}$$
$$+ \sum_{i=1}^{d_r} C_r \cdot (\delta_r^{\text{soft,low}}[i] + \delta_r^{\text{soft,up}}[i]) + \sum_{j=1}^{d_c} C_c \cdot \delta_c^{\text{soft,up}}[j]$$

subject to

$$\pi_t(a|s) \geq 0 \quad \forall a \in \mathcal{A}, s \in \mathcal{S}, t \geq 0$$
$$\pi_t(a|s) = \pi_{t'}(a|s) \quad \forall a \in \mathcal{A}, s \in \mathcal{S}, t, t' \geq 0$$

Here, $\boldsymbol{\alpha}^{\text{low}}, \boldsymbol{\alpha}^{\text{up}}, \boldsymbol{\rho}^{low}, \boldsymbol{\rho}^{up} \in \mathbb{R}^{d_r}$, and $\boldsymbol{\beta}, \boldsymbol{\sigma} \in \mathbb{R}^{d_c}$. We also have non-negativity constraints on the dual variables: $\boldsymbol{\alpha}^{\text{low}}, \boldsymbol{\alpha}^{\text{up}}, \boldsymbol{\beta}, \boldsymbol{\rho}^{\text{low}}, \boldsymbol{\rho}^{\text{up}}, \boldsymbol{\sigma} \geq 0$. A few additional notes:

- For convenience, we will denote the group of dual variables as $\boldsymbol{\lambda} := \{\boldsymbol{\alpha}^{\text{low}}, \boldsymbol{\alpha}^{\text{up}}, \boldsymbol{\beta}, \boldsymbol{\rho}^{\text{low}}, \boldsymbol{\rho}^{\text{up}}, \boldsymbol{\sigma}\}$

- The reward parameter $\boldsymbol{w_\lambda} = [(\boldsymbol{\alpha}^{\text{low}} - \boldsymbol{\alpha}^{\text{up}})^{\dagger}, -\boldsymbol{\beta}^{\dagger}]^{\dagger}$ is used to define the learner's reward function $R_{\boldsymbol{\lambda}}(s) = \langle \boldsymbol{w_\lambda}, \phi(s) \rangle$.

- $\dagger$ is the transpose operator, defined for vectors.

## E.3 Parametric form of the policy

For a given, $\boldsymbol{\lambda} := \{\boldsymbol{\alpha}^{\text{low}}, \boldsymbol{\alpha}^{\text{up}}, \boldsymbol{\beta}, \boldsymbol{\rho}^{\text{low}}, \boldsymbol{\rho}^{\text{up}}, \boldsymbol{\sigma}\}$, the optimal policy $\pi_{\boldsymbol{\lambda}}^{\text{soft}}(a|s)$ is given by

$$\pi_{\boldsymbol{\lambda}}^{\text{soft}}(a|s) = \frac{\exp(Q_{\boldsymbol{\lambda}}^{\text{soft}}(s,a))}{\exp(V_{\boldsymbol{\lambda}}^{\text{soft}}(s))}$$

where the quantities are defined recursively as follows:

$$Q_{\boldsymbol{\lambda}}^{\text{soft}}(s,a) = (\boldsymbol{\alpha}_{low} - \boldsymbol{\alpha}_{up})^{\dagger} \mu_r(\pi_{\boldsymbol{\lambda}}^{\text{soft}}(a|s)) - \boldsymbol{\beta}^{\dagger} \mu_c(\pi_{\boldsymbol{\lambda}}^{\text{soft}}(a|s)) + \gamma \sum_{s' \in \mathcal{S}} T(s'|s,a) V_{\boldsymbol{\lambda}}^{\text{soft}}(s')$$

$$V_{\boldsymbol{\lambda}}^{\text{soft}}(s) = \log \big( \sum_{a \in \mathcal{A}} \exp(Q_{\boldsymbol{\lambda}}^{\text{soft}}(s,a)) \big)$$

This is shown by taking the derivative of the Lagrangian, $\mathcal{L}(\boldsymbol{\pi}, \boldsymbol{\lambda}, \boldsymbol{\psi})$ w.r.t the primal variables, $\pi_t$ and equating it to 0. i.e.

$$\frac{\partial L(\{\pi_t\}_{t=0}^{\infty}, \boldsymbol{\lambda}, \boldsymbol{\psi})}{\partial \pi_t} = 0$$

## E.4 Updated Lagrangian

We find the partial derivatives of the Lagrangian defined in Section E.2 w.r.t all the primal variables, $\delta_r^{\text{soft,low}}, \delta_r^{\text{soft,up}}, \delta_c^{\text{soft,up}}$:

$$\frac{\partial \mathcal{L}}{\partial \delta_r^{\text{soft,low}}[i]} = 0$$
$$\Rightarrow \alpha^{\text{low}}[i] = C_r - \rho^{\text{low}}[i]$$
$$\text{Also, } \frac{\partial \mathcal{L}}{\partial \delta_r^{\text{soft,up}}} = 0$$
$$\Rightarrow \alpha^{\text{up}}[i] = C_r - \rho^{\text{up}}[i]$$
$$\text{And, } \frac{\partial \mathcal{L}}{\partial \delta_c^{\text{soft,up}}} = 0$$
$$\Rightarrow \beta[i] = C_r - \sigma[i]$$

The dual variables satisfy $\boldsymbol{\sigma}, \boldsymbol{\rho}^{\text{low}}, \boldsymbol{\rho}^{\text{up}} \geq 0$. Hence, the above conditions translate into the following constraints on the set of dual variables, $\boldsymbol{\alpha}^{\text{low}}, \boldsymbol{\alpha}^{\text{up}}, \boldsymbol{\beta}$:

$$0 \leq \alpha^{\text{low}}[i] \leq C_r \quad \forall i \in \{1, 2, \ldots, d_r\}$$
$$0 \leq \alpha^{\text{up}}[i] \leq C_r \quad \forall i \in \{1, 2, \ldots, d_r\}$$
$$0 \leq \beta[j] \leq C_c \quad \forall j \in \{1, 2, \ldots, d_c\}$$

The updated Lagrangian now has these additional constraints and is given by:

$$
\begin{aligned}
\mathcal{L}(\boldsymbol{\pi}, \delta_r^{\text{soft,low}}, \delta_r^{\text{soft,up}}, \delta_c^{\text{soft,up}}, \boldsymbol{\lambda}, \boldsymbol{\psi}) = &-H^{\gamma}(A_{0:\infty}, S_{0:\infty}) + (\boldsymbol{\alpha}^{\text{low}} - \boldsymbol{\alpha}^{\text{up}})^{\dagger}(\hat{\mu}_r(\Xi^{\mathsf{T}}) - \mu_r(\pi_t)) + \boldsymbol{\beta}^{\dagger}\mu_c(\pi_t) \\
&+ \sum_{s,t} \psi_{s,t}\left(1 - \sum_{a \in \mathcal{A}} \pi_t(a|s)\right) - (\boldsymbol{\alpha}^{\text{low}})^{\dagger}\delta_r^{\text{soft,low}} - (\boldsymbol{\alpha}^{up})^{\dagger}\delta_r^{\text{soft,up}} \\
&- \boldsymbol{\beta}^{\dagger}\delta_c^{\text{soft,up}} - \boldsymbol{\beta}^{\dagger}\delta_c^{\text{hard}} \\
&- (\boldsymbol{\rho}^{\text{low}})^{\dagger}\delta_r^{\text{soft,low}} - (\boldsymbol{\rho}^{\text{up}})^{\dagger}\delta_r^{\text{soft,up}} \\
&- \boldsymbol{\sigma}^{\dagger}\delta_c^{\text{soft,up}} \\
&+ \sum_{i=1}^{d_r} C_r \cdot (\delta_r^{\text{soft,low}}[i] + \delta_r^{\text{soft,up}}[i]) + \sum_{j=1}^{d_c} C_c \cdot \delta_c^{\text{soft,up}}[j]
\end{aligned}
$$

subject to

$$\pi_t(a|s) \geq 0 \quad \forall a \in \mathcal{A}, s \in \mathcal{S}, t \geq 0$$
$$\pi_t(a|s) = \pi_{t'}(a|s) \quad \forall a \in \mathcal{A}, s \in \mathcal{S}, t, t^{'} \geq 0$$
$$0 \leq \alpha^{\text{low}}[i] \leq C_r \quad \forall i \in \{1, 2, \ldots, d_r\}$$
$$0 \leq \alpha^{\text{up}}[i] \leq C_r \quad \forall i \in \{1, 2, \ldots, d_r\}$$
$$0 \leq \beta[j] \leq C_c \quad \forall j \in \{1, 2, \ldots, d_c\}$$

The set of dual variables becomes $\boldsymbol{\lambda} := \{\boldsymbol{\alpha}^{\text{low}}, \boldsymbol{\alpha}^{\text{up}}, \boldsymbol{\beta}\}$ and $\boldsymbol{\psi} = \{\psi_{s,t}\}_{\forall s_t}$.

## E.5 Dual problem

For any given $\boldsymbol{\lambda}, \boldsymbol{\psi}$, let $g(\boldsymbol{\lambda}, \boldsymbol{\psi})$ be the optimal value for the Lagrangian relaxation problem. Strong Duality holds for both our primal and dual formulations, and the dual optimal policy is also optimal for the primal formulation. Hence, we solve the *concave* dual problem, given by

$$\underset{\boldsymbol{\alpha}^{\text{low}}, \boldsymbol{\alpha}^{\text{up}} \in \mathbb{R}^{d_r}, \boldsymbol{\beta} \in \mathbb{R}^{d_c}, \psi_{s,t} \in \mathbb{R}}{\text{maximize}} g(\boldsymbol{\lambda}, \boldsymbol{\psi})$$

subject to

$$0 \leq \boldsymbol{\alpha}^{\text{low}} \leq C_r$$
$$0 \leq \boldsymbol{\alpha}^{\text{up}} \leq C_r$$
$$0 \leq \boldsymbol{\beta} \leq C_c$$

where $\boldsymbol{\lambda} := \{\boldsymbol{\alpha}^{\text{low}}, \boldsymbol{\alpha}^{\text{up}}, \boldsymbol{\beta}\}$.

## E.6 Gradients for the dual problem

As the dual problem is concave, it can be solved using gradient ascent.
Note that,

$$\nabla_{\psi_{s,t}} g = 1 - \sum_{a \in \mathcal{A}} \pi_{\boldsymbol{\lambda}}^{\text{soft}}(a|s)$$

Here $\pi_{\boldsymbol{\lambda}}^{\text{soft}}$ is the parametric softmax policy described above. This condition is automatically satisfied because $\pi_{\boldsymbol{\lambda}}^{\text{soft}}$ is a probability distribution. For the remaining dual variables, we have the following gradients:

$$\nabla_{\boldsymbol{\alpha}^{\text{low}}} g = \hat{\mu}_r(\Xi^{\mathsf{T}}) - \mu_r(\pi_{\boldsymbol{\lambda}}^{\text{soft}})$$

$$\nabla_{\boldsymbol{\alpha}^{\text{up}}} \quad g = \mu_r(\pi_{\boldsymbol{\lambda}}^{\text{soft}}) - \hat{\mu}_r(\Xi^{\mathsf{T}})$$
$$\nabla_{\boldsymbol{\beta}} \, g = \mu_c(\pi_{\boldsymbol{\lambda}}^{\text{soft}})$$

The (projected) gradient update rules to compute the optimal value of the dual variables $(\boldsymbol{\alpha}^{\text{low}}, \boldsymbol{\alpha}^{\text{up}}, \boldsymbol{\beta})$ are given by the following:

$$\boldsymbol{\alpha}_{\text{next}}^{\text{low}} \leftarrow \boldsymbol{\alpha}^{\text{low}} - \eta \cdot (\mu_r(\pi_{\boldsymbol{\lambda}}^{\text{soft}}) - \hat{\mu}_r(\Xi^{\mathsf{T}}))$$
$$\alpha_{\text{next}}^{\text{low}}[i] \leftarrow \max(0, \alpha_{\text{next}}^{\text{low}}[i]) \quad \forall i \in \{1, 2, \ldots, d_r\}$$
$$\alpha_{\text{next}}^{\text{low}}[i] \leftarrow \min(C_r, \alpha_{\text{next}}^{\text{low}}[i]) \quad \forall i \in \{1, 2, \ldots, d_r\}$$

$$\boldsymbol{\alpha}_{\text{next}}^{\text{up}} \leftarrow \boldsymbol{\alpha}^{\text{up}} - \eta \cdot (\hat{\mu}_r(\Xi^{\mathsf{T}}) - \mu_r(\pi_{\boldsymbol{\lambda}}^{\text{soft}}))$$
$$\alpha_{\text{next}}^{\text{up}}[i] \leftarrow \max(0, \alpha_{\text{next}}^{\text{up}}[i]) \quad \forall i \in \{1, 2, \ldots, d_r\}$$
$$\alpha_{\text{next}}^{\text{up}}[i] \leftarrow \min(C_r, \alpha_{\text{next}}^{\text{up}}[i]) \quad \forall i \in \{1, 2, \ldots, d_r\}$$

$$\boldsymbol{\beta}_{\text{next}} \leftarrow \boldsymbol{\beta} - \eta \cdot (-\mu_c(\pi_{\boldsymbol{\lambda}}^{\text{soft}}))$$
$$\beta_{\text{next}}[j] \leftarrow \max(0, \beta_{\text{next}}[j]) \quad \forall j \in \{1, 2, \ldots, d_c\}$$
$$\beta_{\text{next}[j]} \leftarrow \min(C_c, \beta_{\text{next}}[j]) \quad \forall j \in \{1, 2, \ldots, d_c\}$$

where $\eta$ is the learning rate.

## F  LP Formulation for the Teacher AWARE-CMDP (Section 4.1)

The problem of finding optimal learner-aware teaching demonstrations for the learner in Section 3.1 with linear preferences can be formulated as the following linear program (based on the linear programming formulation for solving MDPs [De, 1960]):

$$\max_z \quad \sum_s \sum_a z(s, a) \langle \mathbf{w}_r^*, \phi_r(s) \rangle \tag{19}$$

$$\text{s.t.} \quad \sum_a z(s', a) = (1 - \gamma) P_0(s') + \gamma \sum_s \sum_a T(s'|s, a) z(s, a) \quad \forall s' \tag{20}$$

$$z(s, a) \geq 0 \quad \forall s, a \tag{21}$$

$$\sum_s \sum_a z(s, a) \phi_c(s)[j] \leq \delta_c^{\text{hard}}[j] \quad \forall j \in \{1, 2, \ldots, d_c\} \tag{22}$$

Here $z$ is a vector of discounted state-action frequencies and $z(s, a)$ refers to state-action frequency for state $s$ and action $a$. The constraints in (22) are the linear preference constraints. From the optimal solution of the LP, an optimal stochastic policy can be extracted by

$$\pi(s, a) := \frac{z(s, a)}{\sum_{a'} z(s, a')}. \tag{23}$$

## G  Bi-Level Optimization Approach (Section 4.2)

We only show the formalism for the most general bi-level problem for learners with linear preferences.

### G.1  Using Dual (discounted) MCE-IRL formulation for the learner model in Section 3.2

The basic bi-level optimization problem that we aim to solve is the following:

$$\max_{\pi^{\mathsf{T}}} \quad R(\pi^{\mathsf{L}})$$
$$\text{subject to} \quad \pi^{\mathsf{L}} \in \arg\max_{\pi} \text{IRL}(\pi, \mu(\pi^{\mathsf{T}})).$$

We will replace the lower-level problem, i.e., $\arg\max_\pi \mathrm{IRL}(\pi, \mu(\pi^\mathsf{T}))$ with its Karush-Kuhn-Tucker conditions [Boyd and Vandenberghe, 2004, Sinha et al., 2018]. The lower-level problem in its dual formulation is given in Appendix E.5.

Omitting details and replacing $R(\pi_\lambda) := \langle \mathbf{w}_r^*, \mu_r(\pi_\lambda)\rangle$, this yields problems of the following form:

$$\max_{\boldsymbol\lambda} \quad \langle \mathbf{w}_r^*, \mu_r(\pi_\lambda)\rangle$$

subject to:

$$0 \leq \boldsymbol\alpha^{\text{low}} \leq C_r$$
$$0 \leq \boldsymbol\alpha^{\text{up}} \leq C_r$$
$$0 \leq \boldsymbol\beta \leq C_c$$
$$\mu_c(\pi_\lambda) \leq (\geq)\delta_c^{\text{hard}}$$

where $\boldsymbol\lambda := \{\boldsymbol\alpha^{\text{low}}, \boldsymbol\alpha^{\text{up}}, \boldsymbol\beta\}$. Here $\pi_\lambda$ corresponds to a *softmax* policy with a reward function $R_\lambda(s) = \langle \boldsymbol{w}_\lambda, \phi(s)\rangle$ for $\boldsymbol{w}_\lambda = [(\boldsymbol\alpha^{\text{low}} - \boldsymbol\alpha^{\text{up}})^\dagger, -\boldsymbol\beta^\dagger]^\dagger$. Thus, finding optimal demonstrations means optimization over *softmax* teaching policies while respecting the learner's preferences.

### G.1.1 Optimal solution

The cases of the above problem we can observe have to be solved separately and the best solution must be picked. That is, we find the following two solutions: (step i) $\boldsymbol\lambda_1^*$, and (step ii) $\boldsymbol\lambda_2^*$. Then pick the best $\boldsymbol\lambda^*$ in (step iii):

**Step i: $\boldsymbol\lambda_1^*$** Compute optimal parameters $\boldsymbol\lambda_1^*$ by solving the following problem:

$$\max_{\lambda} \quad \langle \mathbf{w}_r^*, \mu_r(\pi_\lambda)\rangle$$

subject to:

$$0 \leq \boldsymbol\alpha^{\text{low}} \leq C_r$$
$$0 \leq \boldsymbol\alpha^{\text{up}} \leq C_r$$
$$0 \leq \boldsymbol\beta \leq C_c$$
$$\mu_c(\pi_\lambda) \leq \delta_c^{\text{hard}}$$

**Step ii: $\boldsymbol\lambda_2^*$** Compute optimal parameters $\boldsymbol\lambda_2^*$ by solving the following problem:

$$\max_{\boldsymbol\lambda} \quad \langle \mathbf{w}_r^*, \mu_r(\pi_\lambda)\rangle \tag{24}$$

$$\text{subject to:} \tag{25}$$

$$0 \leq \boldsymbol\alpha^{\text{low}} \leq C_r \tag{26}$$
$$0 \leq \boldsymbol\alpha^{\text{up}} \leq C_r \tag{27}$$
$$\boldsymbol\beta = C_c \tag{28}$$
$$\mu_c(\pi_\lambda) \geq \delta_c^{\text{hard}} \tag{29}$$

**Step iii: $\boldsymbol\lambda^*$** Pick the best solution as

$$\boldsymbol\lambda^* = \arg\max_{\boldsymbol\lambda \in \{\boldsymbol\lambda_1^*, \boldsymbol\lambda_2^*\}} \langle \mathbf{w}_r^*, \mu_r(\pi_\lambda)\rangle$$

This provides the optimal policy for the teacher. The teacher then computes feature expectation of this policy and provide it to the learner.

### G.2 Solving the above problem

We adopt a variant of the Frank-Wolfe algorithm [Jaggi, 2013] to solve the problems of the form:

$$\max_{\boldsymbol\lambda} \quad R(\pi_\lambda) := \langle \mathbf{w}_r^*, \mu_r(\pi_\lambda)\rangle \tag{30}$$

$$\text{subject to:} \tag{31}$$

$$0 \leq \boldsymbol{\alpha}^{\text{low}} \leq C_r \tag{32}$$
$$0 \leq \boldsymbol{\alpha}^{\text{up}} \leq C_r \tag{33}$$
$$0 \leq \boldsymbol{\beta} \leq C_c \tag{34}$$
$$\mu_c(\pi_{\boldsymbol{\lambda}}) \leq (\geq)\delta_c^{\text{hard}} \tag{35}$$

In particular, we take the following steps to optimize the teaching policy $\pi_{\boldsymbol{\lambda}}$:

1. *Initialization.* Find a feasible starting point $\boldsymbol{\lambda}_0$

2. *Optimization.* For $t = 1, 2, \dots$
   - Compute the gradient $\boldsymbol{g}_t = [\nabla_{\boldsymbol{\lambda}} R(\pi_{\boldsymbol{\lambda}})](\boldsymbol{\lambda}_{t-1})$ of the objective at $\boldsymbol{\lambda}_{t-1}$. In experiments we approximate the gradient using finite-differences.
   - Linearize the constraints $\mu_c(\pi_{\boldsymbol{\lambda}}) \leq (\geq)\delta_c^{\text{hard}}$ at $\boldsymbol{\lambda}_{t-1}$ as $\boldsymbol{b}_t + \boldsymbol{A}_t(\boldsymbol{\lambda} - \boldsymbol{\lambda}_{t-1}) \leq (\geq)\delta_c^{\text{hard}}$, where $\boldsymbol{b}_t = \mu_c(\pi_{\boldsymbol{\lambda}_{t-1}})$ and $\boldsymbol{A}_t = [\nabla_{\boldsymbol{\lambda}}\mu_c(\pi_{\boldsymbol{\lambda}})](\boldsymbol{\lambda}_{t-1})$. Again, we employ finite-differences to approximate this linearization. Clearly, we can reuse computation from the gradient estimation of the objective here to reduce computational demands.
   - Solve the direction-finding subproblem (a linear problem):

   $$\max_{\boldsymbol{\gamma}} \quad \langle \boldsymbol{\gamma}, \boldsymbol{g}_t \rangle$$

   subject to:

   $$0 \leq \boldsymbol{\alpha}^{\text{low}} \leq C_r$$
   $$0 \leq \boldsymbol{\alpha}^{\text{up}} \leq C_r$$
   $$0 \leq \boldsymbol{\beta} \leq C_c$$
   $$\boldsymbol{b}_t + \boldsymbol{A}_{t-1}(\boldsymbol{\lambda} - \boldsymbol{\lambda}_{t-1}) \leq (\geq)\delta_c^{\text{hard}}$$

   with optimal solution $\boldsymbol{\gamma}_t^*$. Assuming that the linear approximation of the constraints is accurate locally, the directional vector $\boldsymbol{d}_t = \boldsymbol{\gamma}_t^* - \boldsymbol{\lambda}_{t-1}$ is an ascent direction.
   - Perform a line-search from $\boldsymbol{\lambda}_{t-1}$ to $\boldsymbol{\gamma}_t^*$ and let $\boldsymbol{\lambda}_t$ be the point that maximizes the line search.
   - Upon convergence, terminate the For loop.

Upon convergence of the algorithm, the teacher can use the final $\boldsymbol{\lambda}_t$ for teaching.

*Remark.* Observe that the above algorithm would reduce to the standard Frank-Wolfe algorithm with line-search in the case of linear inequalities only.