[Reviews · NeurIPS 2019]

Reviewer 1



This paper formalizes the problem of inverse reinforcement learning in which the learner’s goal is not only to imitate the teacher’s demonstration, but also to satisfy her own preferences and constraints. It analyzes the suboptimality of learner-agnostic teaching, where the teacher gives demonstrations without considering the learner’s preferences. It then proposes a learner-aware teaching algorithm, where the teacher selects demonstrations while accounting for the learner’s preferences. It considers different types of learner models with hard or soft preference constraints. It also develops learner-aware teaching methods for both cases where the teacher has full knowledge of the learner’s constraints or does not know it. The experimental results show that learner-aware teaching can achieve significantly better performance than learner-agnostic teaching when the learner needs to take into account her own preferences in a simple grid-world domain. I think the problem setting considered by the paper is quite interesting and well formalized. I like Figure 1 that clearly shows the problem of learner-agnostic teaching when the learner has preferences and constraints. To the best of my knowledge, the algorithm presented is original. It builds on previous work, but comes up with a new idea for adaptively teaching the learner assuming the teacher does not know the learner model in the RL setting. The paper is generally clear and well-written. It is a kind of notation-heavy. While I think the authors have done a good job at explaining most things, there are still some places that could be improved. Here are my questions and suggestions: - In equation 1, what is exactly $\delta_r^{hard}$ and $\delta_r^{soft}?$ What is $m$? Is it $d_c$? - In equation 2, what is $\delta_r^{soft, low}$ and $\delta_r^{soft, up}? How do you balance the values of $C_{r}$ and $C_{c}$ when using soft preference constraints? It would be interesting to see how different combinations of these two parameters affect the experimental results. - In all the experiments, how are the teacher demonstration data generated? When the human teachers provide demonstrations, they can be suboptimal. Will this affect the result significantly? - Why does AWARE-BIL perform worse as the learner’s constraints increase (shown in Table 1)? Any hypothesis about this? Does this mean the algorithm might not be able to generalize to cases where the learner has a lot of preference constraints? - When evaluating the learner-aware teaching algorithm under unknown constraints, only the simplest setting (two preference features) similar to Figure 2(a) is considered. I would be curious to know how well the proposed method can perform in settings with more preference features such as L3-L5 (or maybe provide some analysis about it in the paper). - In Figure 3(b), what are the standard errors? Are the reward differences between different algorithms significant? - While the paper focuses on enabling the teacher to optimize the demonstrations for the learner when the learner preferences are known or unknown to the teacher, it would be interesting to address the problem from the learner’s perspective too (e.g., how to learn more efficiently from the given set of demonstrations when the learner has preferences?). UPDATE: Thanks for the author's response! It addresses most of my questions well. I have decided to maintain the same score as it is already relatively high and the rebuttal does not address the significance and scalability issues well enough to cause me to increase my score.

Reviewer 2



Review unchanged after rebuttal. My comments are high level: In the introduction author's may want to reconsider example they use to demonstrate learner's domain conflict. This example of auto-pilot makes one believe that the teacher is the bad one. Teacher suggests that learner should break rules endanger humans etc. to achieve some \pi*. Looks contrived and actually does disservice to the message paper wants to put across. Though answered and addressed later in section 5, learner agnostic teacher should obviously under-perform vs leaner aware due to lack of information [ assuming you can actually use that additional information], this makes results of the paper sound trivial. Again this undersells paper due to the way the authors propose and setup problem in introduction. Only after reading section 5 I realised that the teacher not being aware of exact learner constraints is also handled in the paper. I would suggest this paper being submitted as a journal paper rather than a conference paper, e.g., Theorem 2 and section 5.1 [which are interesting bits of the paper] are entirely in supplementary material. If i decide to read this paper alone without the supplementary material, I would have very little to take away. I am all for writing detailed proofs in Appendix, but when you write algorithms central to your theme in appendix, I start wondering weather this is a journal paper squeezed into 8 page limit due to prestige of NeurIPS.

Reviewer 3



Summary: The paper considers a problem of learning a teacher agent for IL, where a teacher agent aims to find demonstrations that are most informative to an IL agent. Critically, the IL agent has some preferences on its behavior which makes learning from teacher’s optimal demonstrations not appropriate. The paper formulates such an IL agent as maximum causal entropy IRL with preference constraints. The paper then proposes 4 approaches to learn the teacher agent: 2 approaches for known preferences and 2 approaches for unknown preferences. Evaluation on a grid-world task shows that the proposed approaches learn better policy compared to a naïve teacher that is not aware of the IL agent’s constraints. Comments: Overall, this is an interesting paper; It considers an interesting learning setting that is very different from the usual IL setting, and it presents technically sound approaches that come with performance guarantees. However, I am reluctant to give higher score due to limited practicality since the proposed approaches require reward functions to be linear. Also, the experimental evaluation is done in a small grid-world task which is very different from the three examples given in the introduction. ===== I read the rebuttal and the other reviews. My opinion of the paper remains unchanged, and I vote to weak accept the paper.

[Author Response · NeurIPS 2019]

We thank the reviewers for their valuable suggestions. Please find our answers for each reviewer below.

**Reviewer 1**

We thank the reviewer for the positive assessment of our work. Below, we provide a concrete plan of incorporating
reviewer's feedback in the updated version of the paper.

**Extended experimental analysis.** As suggested by the reviewer, we will add a more detailed analysis about the
experimental results in the paper. In particular, we will add the following experiments/details/results: *(i)* evaluation
of learner-aware teaching under unknown constraints for L3-L5 (the findings are similar as for the already presented
experiments); *(ii)* experiments illustrating the effect of $C_r$ and $C_c$ in soft preference constraints; *(iii)* additional details
and discussion of parameter choices in our experiments; *(iv)* reporting the run time of our algorithms, and illustrating
scalability w.r.t. the problem size; and *(v)* reporting standard errors in Figure 3 (b) (the currently reported results in the
paper are significant at significance level 0.1).

**Ideas for outlook and future work.** To the best of our knowledge this paper is the first to consider IRL with preference
constraints. Hence, we primarily focused on developing the theoretical framework and algorithms (for both the known
and unknown constraint settings). Nevertheless, we agree that the directions suggested by the reviewer (more complex
domains and human subject experiments; suboptimal demonstrations and implications on performance; addressing the
problem from a learner's perspective) are important. We will add a discussion on these directions in the revised paper.

**Technical clarifications.** Below we answer the technical questions raised by the reviewer.

• The values $C_r$ and $C_c$ describe a learner's relative importance to mimic the teacher's demonstrations and following
its own preferences, respectively, and are thus properties of a learner and not the parameters of a teacher.
• The performance of AWARE-BIL decreases for increasing learner's constraints because the learner's preferences to
avoid certain cells conflicts with the goal to go to certain cells to accumulate rewards. Note that this decrease is due
to the experimental setup and not due to limitations of AWARE-BIL.
• $\delta_r^{\text{hard}}$ and $\delta_r^{\text{soft}}$ are used to characterize a learner's reward feature matching behaviour as part of the learner's opti-
mization objective: While a mismatch of up to $\delta_r^{\text{hard}}$ between the learner's and teacher's reward feature expectations
incurs no cost regarding the optimization objective, a mismatch larger than $\delta_r^{\text{hard}}$ incurs a cost of $C_r \cdot \|\delta_r^{\text{soft}}\|_p$. Please
also note that $\delta_r^{\text{hard}}$ is a fixed parameter, while $\delta_r^{\text{soft}}$ is an optimization variable. In equation 1, $m$ is the number of
preference constraints of the learner. In general, $m \neq d_c$. Note that we have a typo in the paper in line 108 which
might have caused some confusion: we incorrectly wrote $\delta_c^{\text{soft}} \in \mathbb{R}^{d_c}$ but we wanted to say $\delta_c^{\text{soft}} \in \mathbb{R}^m$. We will
correct this typo and elaborate on the notation in the revised paper.
• $\delta_r^{\text{soft,low}}$ and $\delta_r^{\text{soft,up}}$ are auxiliary variables used to rewrite the constraints on the absolute value of the mismatch in a
form more convenient for optimization. We will add clarification and more details to the revised paper.

**Reviewer 2**

We thank the reviewer for providing useful suggestions and high-level comments on the paper structure.

As suggested, we will remove the auto-pilot example from the introduction and elaborate more on the other two
examples. We will also emphasize that the learner-aware teacher with full-knowledge of the learner allows us to
formalize the problem and introduce a theoretical/algorithmic framework to study the limitations of learner-agnostic
teaching. The real use-case of learner-aware teaching is for incomplete knowledge of the learner. We believe that in this
paper we consider an important new direction for inverse reinforcement learning which we would like to make available
to the community in a timely manner by a conference publication. However, we will revise the paper to include more
details on the algorithms in Section 5.

**Reviewer 3**

We thank the reviewer for appreciating the novelty of the problem setting and providing suggestions for improvements.

**Regarding linearity of the reward function.** It is true that our results are currently for the linear setting. However,
we believe that it is worthwhile to first thoroughly understand this setting. Moreover, as we don't constrain the feature
maps $\phi_r$ and $\phi_c$, the features we consider can be nonlinear functions of a set of "basic" features, which in principle
makes it possible to accommodate quite general situations in our setting. Nevertheless, we agree that a natural next step
is to investigate to what extent our ideas can be extended to nonlinear reward settings.

**Regarding experimental evaluation on more realistic tasks.** Generally, we agree with the reviewer's suggestions
and believe that evaluating our algorithms on more realistic tasks is a natural direction for future work. We would
like to reemphasize that the paper's primary focus is on introducing an important problem setting for IRL, developing
algorithms for the problem, and empirically understanding the performance of these algorithms. We will further extend
the experimental analysis in the paper as outlined in our response to Reviewer 1.

[Meta-Review · NeurIPS 2019]

The paper proposes a really interesting and novel variant of inverse RL with a nice formalization. The proposed algorithms are suitable. While the reviewers felt that the empirical results were weak (lack of scalability and linear reward function limitation), they thought that this was outweighed by the novelty of the problem and the significance of the contribution.